# Nesting of multiple polyhedral plasmonic nanoframes into a single entity

Sungjae Yoo[1,2,7], Jaewon Lee[2,7], Hajir Hilal[2], Insub Jung [2,3], Woongkyu Park [4], Joong Wook Lee[5], Soobong Choi[6] & Sungho Park [2] ✉

The development of plasmonic nanostructures with intricate nanoframe morphologies has attracted considerable interest for improving catalytic and optical properties. However, arranging multiple nanoframes in one nanostructure especially, in a solution phase remains a great challenge. Herein, we show complex nanoparticles by embedding various shapes of three-dimensional polyhedral nanoframes within a single entity through rationally designed synthetic pathways. This synthetic strategy is based on the selective deposition of platinum atoms on high surface energy facets and subsequent growth into solid platonic nanoparticles, followed by the etching of inner Au domains, leaving complex nanoframes. Our synthetic routes are rationally designed and executable on-demand with a high structural controllability. Diverse Au solid nanostructures (octahedra, truncated octahedra, cuboctahedra, and cubes) evolved into complex multi-layered nanoframes with different numbers/shapes/sizes of internal nanoframes. After coating the surface of the nanoframes with plasmonically active metal (like Ag), the materials exhibited highly enhanced electromagnetic near-field focusing embedded within the internal complicated rim architecture.

Three-dimensional (3D) complex nanoframes wherein multiple polyhedral nanoframes are integrated in a confined single entity are analogous to Kepler's cosmic bowl, wherein the distance relationships between the six planets could be represented in terms of the five platonic solids enclosed within a sphere. Structurally, if polyhedral nanoframes are symmetrically arranged in a single entity, it will lead to extraordinary structural synergistic effects like highly efficient trapping of electromagnetic radiation within the sophisticated internal nanocavity. Such a complicated morphology would be ideal for a wide range of applications such as surface-enhanced Raman scattering (SERS)[1], biosensors[2], catalysts[3–5]. Among various morphologies of nanostructures, nanoframes typically have a unique structural feature that can increase the accessibility of every surface in a given space, offering great potential for interactions with light and surface adsorbates. To date, enormous efforts have been devoted to improving the physicochemical properties of nanoframes[6–18] by controlling the structural parameters such as shape, size, thickness, and composition. Nevertheless, all the nanoframes synthesized so-far are structurally simple and mainly composed of single-rim structures, limiting the utilization of inner voids, making it difficult to harness efficient structural coupling effects or light-matter interactions[19–23]. In this regard, integrating multiple nanoframes in a single entity can be considered as a new paradigm for efficiently amplifying the structural characteristics of nanoframes. However, the synthetic strategies for complex nanoframes with different multiple internal geometries are rare, and the synthesis of complex multiple nanoframes in a controllable fashion (especially in a solution phase with a high yields and homogeneity in size and shape) remains a great challenge[24–30]. Herein,

[1]Research Institute for Nano Bio Convergence, Sungkyunkwan University, Suwon 16419, Republic of Korea. [2]Department of Chemistry, Sungkyunkwan University, Suwon 16419, Republic of Korea. [3]Institute of Basic Science, Sungkyunkwan University, Suwon 16419, Republic of Korea. [4]Medical & Bio Photonics Research Center, Korea Photonics Technology Institute (KOPTI), Gwangju 61007, Republic of Korea. [5]Department of Physics and Optoelectronics Convergence Research Center, Chonnam National University, Gwangju 61186, Republic of Korea. [6]Department of Physics, Incheon National University, Incheon 22012, Republic of Korea. [7]These authors contributed equally: Sungjae Yoo, Jaewon Lee. ✉e-mail: spark72@skku.edu

we develop rationally designed on-demand multi-step chemical reactions for complex nanoframes with a high degree of structural controllability, which we denoted such complex nanoframes as N-th nanoframes.

## Results

### Synthetic strategy for N-th nanoframes

A schematic illustration (Fig. 1) of synthetic processes for N-th nanoframes exhibits the detailed multi-step routes to make each complex nanoframe. As a proof of concept, we show three distinctive pathways that are designed using different starting shapes of Au solid nanoparticles (such as octahedra, truncated octahedra, and cubes, enclosed by only {111}, a combination of {111} and {100}, and only {100} facets, respectively) followed by on-demand combinations and multiple repetition of the several distinctive chemical toolkits. Each distinctive chemical reaction is described by arrows of different colors. These are listed as follows: (1) Rim-selective growth of Pt (blue arrows): Pt ions are preferentially reduced at the edge and vertex regions of Au solid nanoparticles. (2) Well-faceted overgrowth of Au (red arrows): Au atoms are deposited over the entire surface, retaining well-defined facets and evolving to platonic and truncated platonic nanoparticles. (3) Selective etching of inner Au (black arrows): Au atoms are selectively etched away, leaving mainly a Pt framework with residual Au. It is worth mentioning that multiple repetition of well-faceted overgrowth of Au step allows one to precisely modulate both the size and shape of Au nanostructure intermediates, resulting in multiple polyhedral nanoframes with tailorable structures. We denoted such nanoframes as N-th nanoframes. To clearly differentiate each structure, we suggest a nomenclature of [Order-Composition-Shape-Morphology]. The order indicates the total number of nanoframes embedded in a single entity (e.g., 1st, 2nd, 3rd, and 4th). Composition implies the main element composing the nanostructure (e.g., Pt, Au, and Ag), and shape depicts each geometry starting from the core to the outer shape (e.g., octahedron (O), truncated octahedron (TO), cuboctahedron (CO), and cube (C)). The last abbreviation indicates if the material is composed of solid nanoparticles (NPs) or nanoframes (NFs).

### Synthesis of 2nd nanoframes

By following the synthetic pathway shown in Fig. 1, we describe how one can synthesize 2nd-Pt-O:O-NFs (Far-left scheme in Fig. 1). First, Au octahedral NPs with sizes of $48 \pm 1$ nm were employed as a starting template (Fig. 2a), and we applied rim-selective growth of Pt

decorating edges and vertexes with Pt (Fig. 2b), well-faceted overgrowth of Au to enlarge octahedral NPs (Fig. 2c), and rim-selective growth of Pt again leading to 2nd-Au@Pt-O-NPs (Fig. 2d). In the first rim-selective growth of Pt step, Ag+ ions were added to the reaction solution in the presence of ascorbic acid, resulting in the formation of thin Ag layer on the whole surface area of Au NPs. Subsequently, Pt atoms were selectively deposited along the edges and vertexes of Au octahedral NPs because the relatively higher surface energy of those protruded sites can facilitate a galvanic replacement reaction between Pt4+ ions and the thin Ag layer, leading to 1st-Au@Pt-O-NPs with a total size of $77 \pm 2$ nm (Fig. 2b). In the well-faceted overgrowth of Au step, the morphologies of 1st-Au@Pt-O-NPs are transformed into solid 2nd-Au-O-NPs ($100 \pm 2$ nm, Fig. 2c). The size of resulting solid 2nd-Au-O-NPs is tunable from $90 \pm 2$ to $111 \pm 4$ nm by simply controlling the amount of Au ions (Supplementary Fig. 1). This well-faceted overgrowth of Au step is an important intermediate step to allow subsequent chemical reaction (e.g., rim-selective growth of Pt) aided by formation of flattened surfaces and sharp edges via the Frank-van der Merwe mode as shown in Fig. 2e. In the nucleation and growth step of Au, Pt sites can act as nucleation sites in the presence of both Ag+ ions and Cl- ions and pre-formed thin Ag layers can induce the epitaxial growth of Au in a lateral direction, enlarging the flat Au terrace along the Au basal plane. To investigate the underlying mechanism of the well-faceted overgrowth of Au step, we controlled the experimental parameters such as counter anions, presence of Ag+ ions, as well as the concentration of Au3+ ions. As shown in Supplementary Fig. 2, three different growth patterns of Au on the surface of 1st-Au@Pt-O-NPs were observed. As the concentration of Au ions increases, Au atoms are preferentially reduced on Pt regions in the presence of both Ag+ ions and Cl- ions (reaction condition 1), forming flattened surfaces and sharp edges via the Frank-van der Merwe (FW) mode. As Au growth proceeds, edge domains expand with flattened facets in symmetric dual directions, and they are eventually merged, resulting in enlarged octahedral Au NPs with eight well-defined flat facets (as shown in Supplementary Fig. 2b–d). In contrast, in the absence of Ag+ ions (reaction condition 2), non-epitaxial growth of Au is observed, leading to Au nanoparticles with roughened surfaces following Stransky–Krastanov (SK) mode, as shown in Supplementary Fig. 2e–g. In addition, when counter-anions changed to Br- ions (reaction condition 3), indistinctively evolved dull edges were observed as shown in Supplementary Fig. 2h–j, indicating that Cl- ions play an important role in inducing the formation of thin Ag layers. This is caused by the higher reduction potential of AgCl complexes (E°/V = 0.22 vs. Ag/AgCl) than that of AgBr complexes (E°/V = 0.07 vs. Ag/AgCl). The presence of a thin layer of Ag is necessary to reduce the difference in lattice constant between Pt and Au (e.g., lattice constant of Au and Pt is 0.4065 and 0.3912 nm, respectively), leading to the epitaxial growth and formation of well-defined facets. As evidenced by EDS image mapping data (Supplementary Fig. 3), Ag atoms are preferentially deposited along the Pt sites, and the atomic percentages of Au, Pt, and Ag were 60, 20, and 20%, respectively. It should be noted that preferential deposition of Ag on the Pt sites results from the higher catalytic activity of the Pt surface compared to that of the Au surface[31,32]. Additionally, the synthetic pathway can be expanded to other different shapes such as truncated octahedral and cube shapes, proving the general applicability of well-faceted overgrowth of Au (Supplementary Fig. 4). After applying the second rim-selective growth of Pt step, 2nd-Au@Pt-O-NPs with total size of $125 \pm 4$ nm could be obtained (Fig. 2d). Bright lines along the edge sites indicate that Pt atoms are selectively deposited on the edge regions. Shape transformation from 1st-Au-O-NPs to 2nd-Au@Pt-O-NPs could be monitored through UV–vis spectrum (Supplementary Fig. 5). After rim-selective growth of Pt, LSPR peaks red-shifted accompanying with peak damping due to enlarging the size of nanoparticle and poor plasmonic activity of Pt in visible regions. In contrast, LSPR peaks blue-shifted accompanying with increased extinction after well-faceted

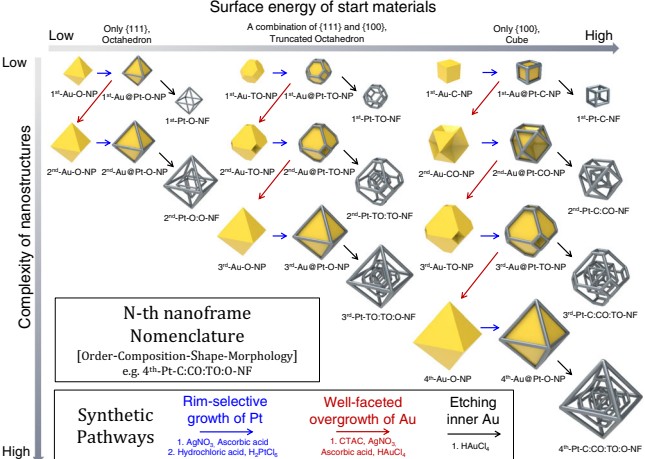

**Fig. 1 | Schematic illustration of multi-step synthetic pathway for N-th nanoframes.** Three distinctive synthetic pathways including rim-selective growth of Pt (blue arrow), well-faceted overgrowth of Au (red arrow) and etching inner Au (black arrow) are represented, and the nomenclature for N-th nanoframes is suggested.

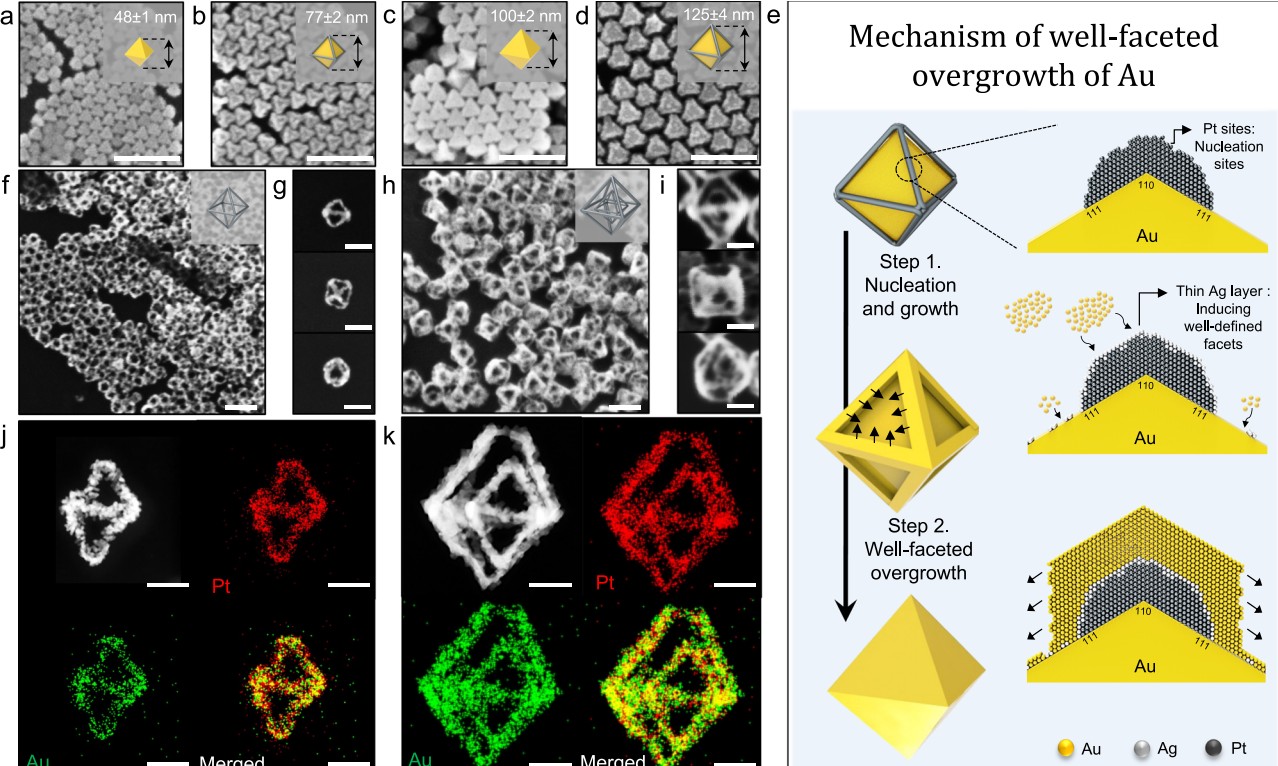

**Fig. 2 | Representative schematic illustration for mechanism of well-faceted overgrowth of Au, and morphology evolution from 1st to 2nd nanoframes.** SEM images of **a** 1st-Au-O-NPs, **b** 1st-Au@Pt-O-NPs, **c** 2nd-Au-O-NPs, and **d** 2nd-Au@Pt-O-NPs, scale bar = 200 nm. **e** Schematic illustration that shows the mechanism of well-faceted overgrowth of Au. Low magnification and zoomed-in SEM images viewed from <110>, <100>, and <111> directions of **f, g** 1st-Pt-O-NFs, and **h, i** 2nd-Pt-O:O-NFs, scale bar = 100 nm. **j, k** STEM images and EDS elemental mapping images of **j** 1st-Pt-O-NFs and **k** 2nd-Pt-O:O-NFs, scale bar = 30 nm.

overgrowth of Au. This is because the resulting nanostructures act as pure Au nanoparticles. Finally, in the first selective etching of Au step, inner bulk Au domains were etched away through the comproportionation reaction of Au+ ions, producing single Pt frameworks that have an octahedral shape and inner vacancy space (denoted as 1st-Pt-O-NFs) as shown in Fig. 2f, g, j. Notably, after etching inner Au domains of 2nd-Au@Pt-O-NPs, Pt nanoframes with dual nanoframes (denoted as 2nd-Pt-O:O-NFs) were obtained as shown in Fig. 2h, i, k. The 3D movie clip obtained with combined tilt series of TEM images taken from −60° to +60° angles (Supplementary Movie 1) demonstrates that 2nd-Pt-O:O-NFs are composed of inner and outer nanoframes both with octahedral shapes, possessing intra-cavities connected via thin metal ligaments. Further, 1st-Pt-O-NFs and 2nd-Pt-O:O-NFs are mainly composed of Pt, and a residual amount of Au confirmed by energy dispersive X-ray spectroscopy (EDS) in scanning transmission electron microscopy (STEM).

## Synthesis of 3rd nanoframes

As shown in the synthetic pathway (middle scheme in Fig. 1), we applied a series of chemical reactions including multiple repetition of well-faceted overgrowth of Au to Au TO NPs bounded by {111} and {100} facets (Fig. 3). Au TO NPs with a total size of 55 ± 1 nm were utilized as a template (Fig. 3a) and applied for rim-selective growth of Pt (1st-Au@Pt-TO-NPs), well-faceted overgrowth of Au (2nd-Au-TO-NPs), then again rim-selective growth of Pt (2nd-Au@Pt-TO-NPs), well-faceted overgrowth of Au (3rd-Au-O-NPs), and finally rim-selective growth of Pt, leading to 3rd-Au@Pt-O-NPs (Fig. 3b–f, respectively). Shape transformation from 1st-Au-TO-NPs to 3rd-Au@Pt-O-NPs could be also monitored through UV-vis spectrum (Supplementary Fig. 6). Notably, in the selective etching of inner Au step, inner Au domains of 1st, 2nd, and 3rd Au@Pt NPs were etched away, producing three kinds of complex nanoframes (denoted as 1st-Pt-TO-NFs (Fig. 3g, h, m), 2nd-Pt-TO:TO-NFs

(Fig. 3i, j, n), and 3rd-Pt-TO:TO:O-NFs (Fig. 3k, l, o)). It is noteworthy that in the first well-faceted overgrowth of Au, as the concentration of Au3+ ions gradually increases, the total size of the resulting structures increased from 90 ± 3 to 137 ± 5 nm and shape of intermediates of Au nanostructure can be precisely tuned from truncated octahedron to octahedron (Supplementary Fig. 7). Thereby, the outer shape of 2nd nanoframes could be controlled from truncated octahedral to octahedral nanoframes and the intra-nanogap distance between the inner and outer nanoframes of 2nd-Pt-TO:O-NFs was precisely tuned (Supplementary Fig. 8). Remarkably, the 3rd-Pt-TO:TO:O-NFs could be obtained wherein three different nanoframes are embedded in a single entity, namely, inner truncated octahedra, a middle truncated octahedral nanoframe, and the outer octahedral nanoframe, retaining intra-nanogaps regions among nanoframes, as shown in Fig. 3k, l. Further, the 3D movie clip obtained with a combined tilt series of TEM images showed the structural details of 3rd-Pt-TO:TO:O-NFs, proving that three different nanoframes are beautifully overlapped in a 3D confined single entity (Supplementary Movie 2).

## Synthesis of 4th nanoframes

Now, we suggest how to synthesize N-th nanoframes with four different kinds of geometrical shapes, which illustrates the ultimate controllability of the resulting N-th nanoframes. The most significant morphology transformation of intermediates was observed when Au cubes enclosed by {100} facets were utilized as a starting material and four different complex nanoframes can be realized. The overall route to the 4th NFs is described in the far-right side of Fig. 1. Au cubes with total sizes of 30 ± 1 nm were employed as a starting material (Fig. 4a). Then, rim-selective growth of Pt and well-faceted overgrowth of Au were adopted multiple times. In the first rim-selective growth of Pt step, Pt atoms were selectively deposited on the edges of Au cubes, leading to 1st-Au@Pt-C-NPs (Fig. 4b). Subsequently, as the well-faceted

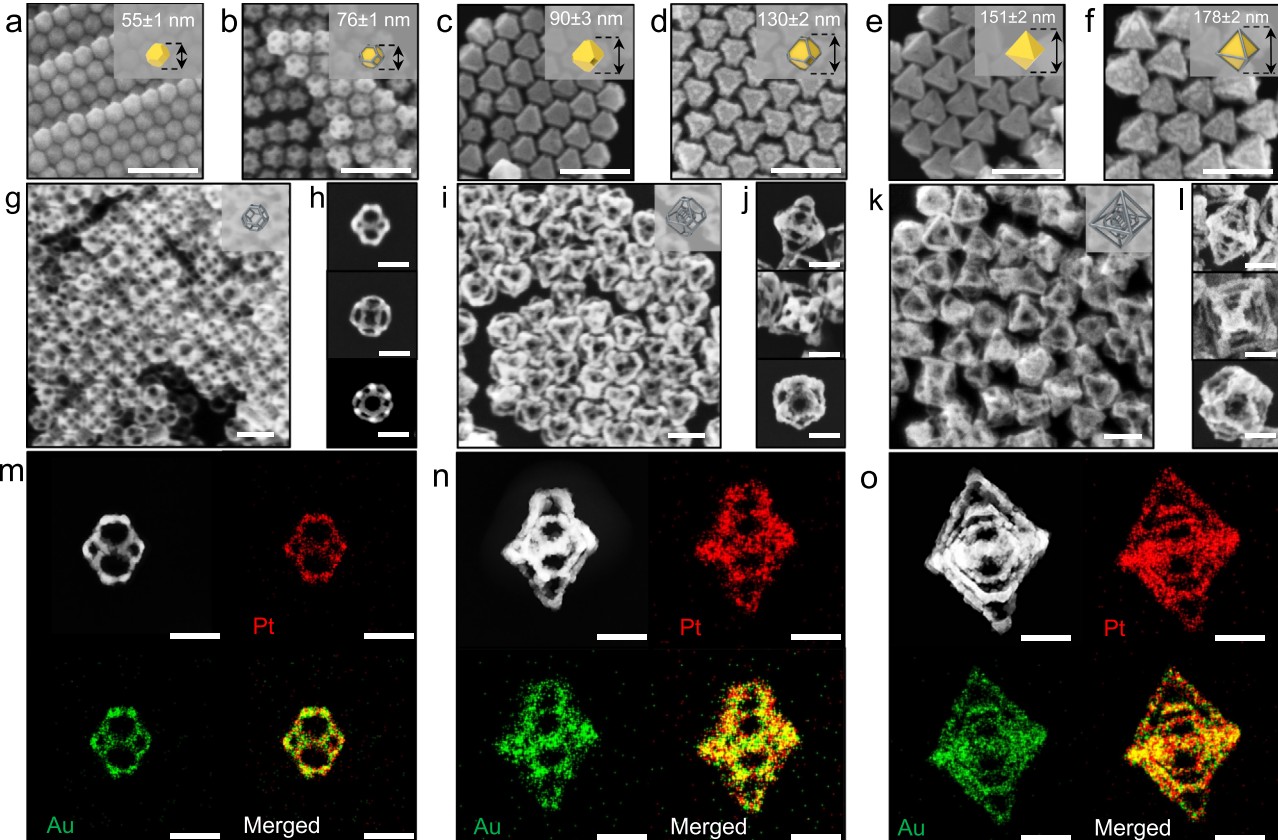

**Fig. 3 | Morphology evolution from 1st to 3rd nanoframes.** SEM images of **a** 1st-Au-TO-NPs, **b** 1st-Au@Pt-TO-NPs, **c** 2nd-Au-TO-NPs, **d** 2nd-Au@Pt-TO-NPs, **e** 3rd-Au-O-NPs, and **f** 3rd-Au@Pt-O-NPs, scale bar = 200 nm. **g–l** Low magnification and zoomed-in SEM images viewed from <110>, <100>, and <111> directions of **g**, **h** 1st-Pt-TO-NFs, **i**, **j** 2nd-Pt-TO:O-NFs, **k**, **l** 3rd-Pt-TO:O:O-NFs, scale bar = 100 nm. STEM images and EDS elemental mapping images of **m** 1st-Pt-TO-NFs, **n** 2nd-Pt-TO:TO-NFs, and **o** 3rd-Pt-TO:TO:O-NFs, scale bar = 50 nm.

overgrowth of Au and rim-selective growth of Pt steps were applied repeatedly, both the shape and total size of intermediates of the Au nanostructure can be precisely modulated from 1st-Au@Pt-C-NPs with sizes of $40 \pm 1$ nm to Au cuboctahedra with sizes of $70 \pm 1$ nm (Fig. 4c, 2nd-Au-CO-NPs), Au truncated octahedra with a size of $98 \pm 4$ nm (Fig. 4e, 3rd-Au-TO-NPs), and Au octahedra with sizes of $136 \pm 3$ nm (Fig. 4g, 4th-Au-O-NPs). Shape transformation from 1st-Au-C-NPs to 4th-Au@Pt-O-NPs could be also monitored through UV–vis spectrum (Supplementary Fig. 9). Notably, as well-faceted overgrowth of Au step proceeded, the shape transformed from cubes into cuboctahedra, truncated octahedra, and octahedra. Accordingly, the size evolved from $72 \pm 2$ to $142 \pm 3$ nm, as shown in Supplementary Fig. 10b–h. Specifically, as Au overgrowth of 1st-Au@Pt-C-NPs gradually increased, eight vertex regions and six terrace regions were transformed into eight terrace regions and six vertex regions, respectively, leading to perfect octahedra (Supplementary Fig. 10). In the selective etching of Au step, inner bulk Au domains of the resulting 1st, 2nd, 3rd, and 4th Au@Pt NPs (Fig. 4b, d, f, h) were etched away to Au+ ions, producing complex nanoframes (denoted as 1st-Pt-C-NFs (Fig. 4i, j, q), 2nd-Pt-C:CO-NFs (Fig. 4k, l, r), 3rd-Pt-C:CO:TO-NFs (Fig. 4m, n, s), and 4th-Pt-C:CO:TO:O-NFs (Fig. 4o, p, t), respectively). In addition, the outer shape of the double nanoframe could be controlled by tuning the degree of well-faceted overgrowth of Au, leading to 2nd-Pt-C:TO-NFs and 2nd-Pt-C:O-NFs (Supplementary Fig. 11). Remarkably, 4th-Pt-C:CO:TO:O-NFs were successfully obtained, wherein nanoframes with four different geometrical morphologies and sizes were integrated into a single entity, amplifying the controllability of our synthetic strategies. Again, TEM tomography 3D movie clip measured from −60° to +60° angles (Supplementary Movie 3) remarkably demonstrates

that small cube nanoframes are located in the center of the whole structures. Similarly, cuboctahedral, truncated octahedral, and octahedral nanoframes are contained in one 3D entity. It is noteworthy that low magnification SEM images (Fig. 4i, k, m, o) show a high degree of homogeneity in both shape and size, demonstrating the high controllability of our multi-step synthetic strategy.

**Investigating the localized surface plasmon resonance (LSPR) properties of N-th nanoframes**

To trap visible light inside N-th nanoframes, Ag atoms are deposited on the Pt scaffolds, leading to complex Ag nanoframes with different numbers of inner nanoframes in a single entity (Supplementary Fig. 12). EDS analysis data show that a thin Ag layer was deposited on the entire surface while keeping their structural complexity (Supplementary Fig. 13). Typically, through the ICP-MS analysis, Ag atomic fractions of 1st, 2nd, 3rd, and 4th of Ag nanoframes were found to be 22, 20, 19, and 20%, respectively (Supplementary Table 1). The corresponding LSPR profiles of 1st, 2nd, 3rd, and 4th Ag nanoframes were obtained by ultraviolet–visible-near-infrared (UV–Vis-NIR) spectroscopy as shown in Fig. 5a. Dipole mode of 1st Ag nanoframes blue-shifted from 936 to 534 nm resulting from surface plasmon coupling between inner and outer nanoframe. Then, as the number of nanoframes increased from double to quadruple, the dipole mode of nanoframes red-shifted from 534 to 564 nm and 620 nm accompanying with peak broadening. This trend was well agreed with theoretically calculated UV–vis-NIR spectrum obtained by theoretical simulations based on a finite-element method (FEM) as shown in Supplementary Fig. 14. Charge distribution shown in Fig. 5d (red: electron rich, blue: electron deficient) clearly indicates that the surface charge density

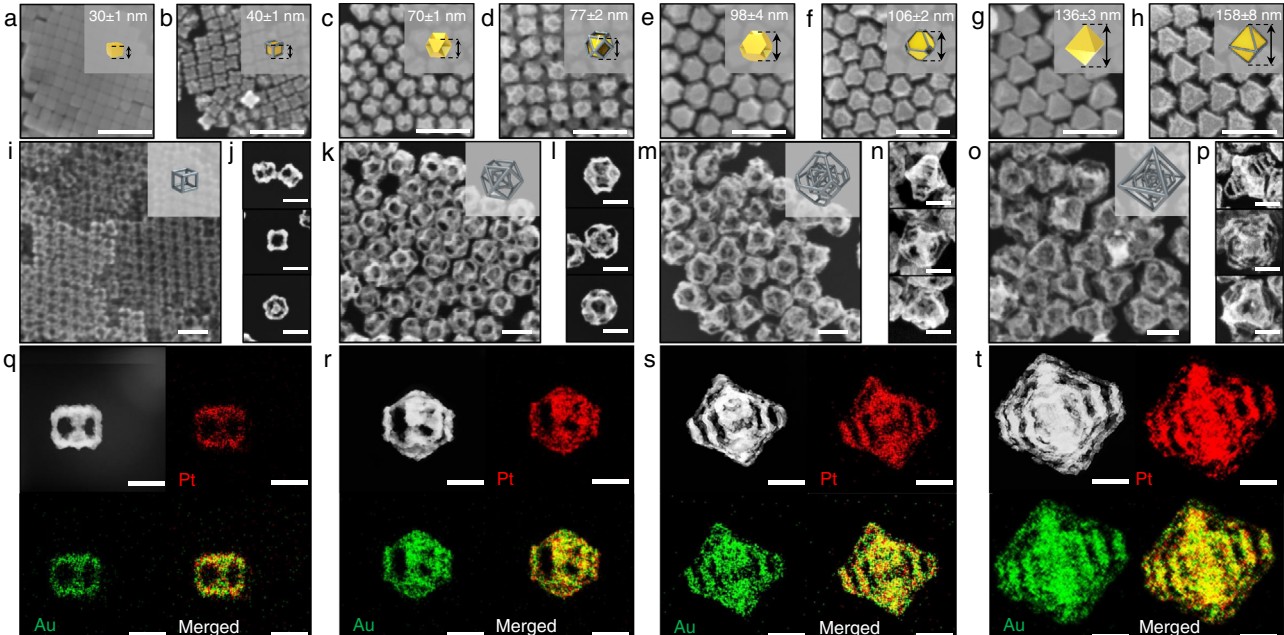

**Fig. 4 | Morphology evolution from 1st to 4th nanoframes.** SEM images of **a** 1st-Au-C-NPs, **b** 1st-Au@Pt-C-NPs, **c** 2nd-Au-CO-NPs, **d** 2nd-Au@Pt-CO-NPs, **e** 3rd-Au-TO-NPs, **f** 3rd-Au@Pt-TO-NPs, **g** 4th-Au-O-NPs, and **h** 4th-Au@Pt-O-NPs, scale bar = 200 nm. **i–p** Low magnification and zoomed-in SEM images viewed from <110>, <100>, and <111> directions of **i** and **j** 1st-Pt-C-NFs, **k** and **l** 2nd-Pt-C:CO-NFs, **m**, **n** 3rd-Pt-C:CO:TO-NFs, **o**, **p** 4th-Pt-C:CO:TO:O-NFs, scale bar = 100 nm. STEM image and EDS elemental mapping images of **q** 1st-Pt-C-NFs, **r** 2nd-Pt-C:CO-NFs, **s** 3rd-Pt-C:CO:TO-NFs, and **t** 4th-Pt-C:CO:TO:O-NFs, scale bar = 50 nm.

significantly increased in a single entity as the number of nanoframes increased from single to quadruple, which is a distinctive characteristic of N-th nanoframes. We expect that surface plasmon coupling would occur in a complex manner, representing a broad LSPR feature (Fig. 5a). Furthermore, we obtained the single-particle surface-enhanced Raman scattering (spSERS) spectra as shown in Fig. 5b. We used 2-naphthalenethiol as an analyte for SERS measurements and a 633 nm laser (Laser power: 170 µW) was used. Position of the individual N-th nanoframes was confirmed by Rayleigh scattering and SEM images (Supplementary Fig. 15). SERS signals were obtained at different excitation wavelength, indicating that the highest intensity of Raman signals with relatively high reproducibility SERS signals was observed at the 633 nm excitation wavelength (Supplementary Figs. 16, 17, and 18). Remarkably, the representative spSERS signals (at 633 nm excitation wavelength) show that the intensity of Raman signals is exponentially amplified as the number of inner nanoframes increases from singular- to quadruple-nanoframes in a single entity. (Fig. 5b, c). Calculated enhancement factors (at 633 nm) are $6.3 \times 10^7$ (for 1st Ag nanoframes), $3.7 \times 10^8$ (for 2nd Ag nanoframes), $7.6 \times 10^8$ (for 3rd Ag nanoframes), and $1.3 \times 10^9$ (for 4th Ag nanoframes) (Supplementary Fig. 19). Among these, the enhancement factor of 4th Ag nanoframes belongs to the highest ranking compared to that of previously plasmonic materials (Supplementary Table 2). To systematically verify surface plasmon coupling phenomena of complex Ag nanoframes with varying numbers of nanoframes, electromagnetic near-field enhancement data were obtained by FEM (Fig. 5e). As shown in FEM field distribution images (Fig. 5e and Supplementary Fig. 20), 4th Ag nanoframes maximize the near-field focusing on those interstitial regions among nanoframes, wherein the adsorbed analytes would experience the strongly confined near-field, leading to highly amplified spSERS signal. As shown in Fig. 5c, the increased SERS signals were well correlated with the integrated field enhancement within N-th nanoframes. The synthetic strategy for N-th nanoframes has a high structural controllability within one entity. The resulting morphology consists of overlapped nanoframes with each different shape, which is in between the open frame structure and closed shell structure systems. It is feasible to utilize the inner nanostructures for further enhancing their corresponding optical and physical properties, which differs from other similar systems (e.g., nanorattles, nanocages, and nanomatryoshkas). Although we show one example at the current stage, their near-field focusing as a function of the number of wrapping nanoframes, we believe that there are many other related applications by using N-th nanoframes, especially wherein the inner morphology control, penetration depth of light, and mass-transportation of analytes are important.

Kepler tried to mimic the structure of the universe by ordering various solids (octahedron, icosahedron, dodecahedron, tetrahedron, and cube) in one entity. Likewise, N-th nanoframes allow multiple nanoframes of different geometries in a controlled way and mimic such a complicated arrangement in one nanoparticle. The synthetic chemical tools, executable on-demand in a customized way, lead to a high degree of homogeneity in both shape and size. The void space consists of many intertwined but separated nanorims, wherein the near-field focusing could be maximized, as verified by both spSERS measurements and theoretical calculations.

## Methods
### Materials and Instruments
**Materials.** Hydrogen tetrachloroaurate (III) hydrate (HAuCl$_4$·nH$_2$O, 99%) and hydrogen hexachloroplatinate (IV) hydrate (H$_2$PtCl$_6$·nH$_2$O, 99%) were purchased from Kojima. Sodium borohydride (NaBH$_4$, 98%) and silver nitrate (AgNO$_3$, 99.8%) were purchased from Junsei. Sodium iodide (NaI, 99.5%) and L-ascorbic acid (AA, C$_6$H$_8$O$_6$, 99.5%) were supplied by Sigma Aldrich. Hydrochloric acid (HCl, 35%), sodium hydroxide (NaOH, 98%), and sodium bromide (NaBr, 99.0%) were purchased from Samchun. Cetyltrimethylammonium bromide (CTAB, C$_{19}$H$_{42}$BrN, 95%) was purchased from Sigma-Aldrich. Cetyltrimethylammonium chloride (CTAC, C$_{19}$H$_{42}$ClN, 95%) was purchased from Tokyo Chemical Industry. All chemical materials were dissolved in distilled water (18.2 MΩ) prepared using a water purification system (Milli-Q, Millipore).

**Instruments.** Field emission scanning electron microscopy (FESEM) images were obtained using JSM-7100F and JSM-7800F instruments

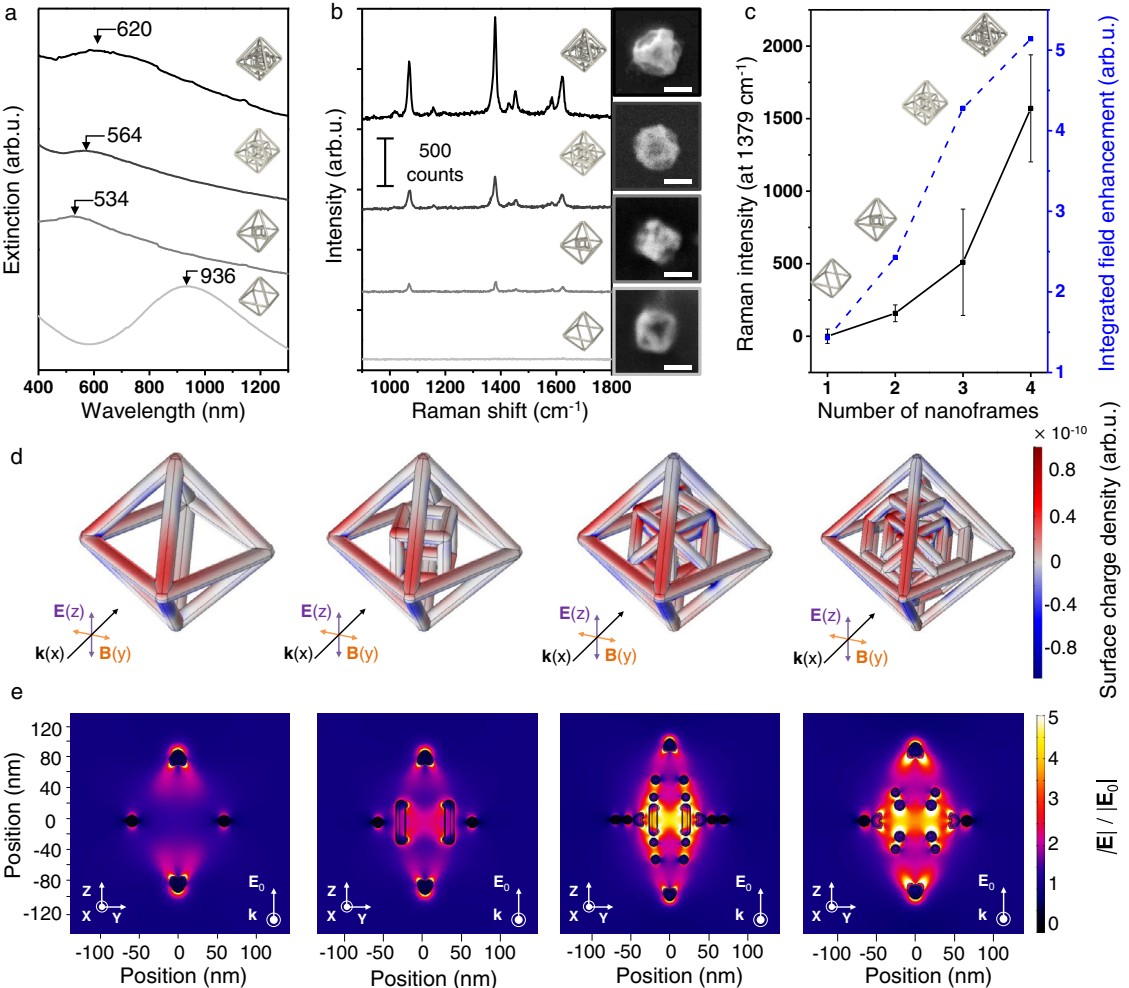

**Fig. 5 | Investigating the optical properties of complex Ag nanoframes with varying the number of nanoframes. a** LSPR band profile of 1st, 2nd, 3rd, and 4th Ag nanoframes, **b** representative single-particle Raman signals of 1st, 2nd, 3rd, and 4th Ag nanoframes originating from their corresponding SEM images (scale bar = 200 nm), **c** plotted data of single-particle Raman intensity (at 1379 cm$^{-1}$) and theoretically calculated values of electric near-field enhancement as function of the number of nanoframes. Error bars indicate the standard deviations of Raman intensities at 1379 cm$^{-1}$ of 1st, 2nd, 3rd, and 4th Ag nanoframes, **d** charge distribution and **e** electromagnetic near-field contour maps (in the YZ cross-section) of 1st, 2nd, 3rd, and 4th Ag nanoframes with different numbers of nanoframes obtained by FEM simulation (Excitation wavelength: 633 nm).

(JEOL). JEM-2100F and JEM-ARM 200F instruments (JEOL) were used to acquire transmission electron microscopy (TEM) images. Vortex shaker (Vortex-Genie 2, Scientific Industries, Inc.) was used to shake the reaction solution thoroughly. UV–vis–NIR absorption spectra were acquired using a spectrophotometer (Shimadzu UV-3600). Raman spectra and images were recorded on a Raman microscope (Ntegra, NT-MDT).

### Synthetic method of nanoparticles

**Synthesis of Au Octahedral Nanoparticles.** Au octahedral nanoparticles were synthesized by a seed-mediated method[21]. To synthesize the Au seed solution, 600 μL of 10 mM of ice-cold NaBH$_4$ solution was injected into a mixture of 7 mL of 75 mM CTAB, 87.5 μL of 20 mM HAuCl$_4$ with vigorous mixing for 3 h. First, 6 mL of seed solution diluted 100 times in deionized water (DIW) was added to a solution containing 480 mL of 16 mM CTAB, 200 μL of 20 mM HAuCl$_4$, and 6 mL of 0.1 M AA. The mixture was kept at 30 °C for 12 h. Next, 210.8 mL of 100 mM CTAB, 1.4 mL of 20 mM HAuCl$_4$, and 6.53 mL of 0.1 M AA were mixed in a round bottom flask, and 140.5 mL of second Au octahedral nanoparticles was added to the flask. The mixture was kept at 30 °C for 4 h. The resulting solution was washed twice by centrifugation (7168 × g, 20 min), and dispersed in DIW (optical density: 1.2, 564 nm).

**Synthesis of Au Truncated Octahedral Nanoparticles.** Au truncated octahedral nanoparticles were synthesized with a slight modification of the above method. First, to synthesize Au seed solution, 600 μL of 10 mM of ice-cold NaBH$_4$ solution was added to a mixture of 7 mL of 50 mM CTAB, 87.5 μL of 20 mM HAuCl$_4$ with vigorous stirring for 3 h. For the first growth process, 0.45 mL of seed solution diluted 100 times was added to a solution containing 12 mL of 16 mM CTAB, 25 μL of 20 mM HAuCl$_4$, and 25 μL of 38.8 mM AA. The mixture was kept at 30 °C for 12 h. For the second growth, 12 mL of the solution from the first growth process was added to a solution containing 12 mL of 16 mM CTAB, 3 mL of 2 mM HAuCl$_4$, and 4.6 mL of 12 mM AA. After 5 min of reaction, the samples were centrifuged at 2580 × g for 30 min. This washing process was repeated two times, and dispersed in DIW (optical density: 1.0, 539 nm).

**Synthesis of Au Cubic Nanoparticles.** Au cubic nanoparticles were synthesized by a seed-mediated method[33]. To synthesize the Au seed solution, 600 μL of 10 mM of ice-cold NaBH$_4$ solution was injected into

a mixture containing 9.75 mL of 100 mM CTAB and 250 μL of 20 mM HAuCl$_4$ with stirring for 3 min. The solution was kept at 30 °C for 3 h. Next, to synthesize 10 nm Au nanosphere, 2 mL of 200 mM CTAC, 2 mL of 0.5 mM HAuCl$_4$, 1.5 mL of 0.1 M AA, and 50 μL of the CTAB-capped seed solution were added sequentially with stirring for 15 min. It was centrifuged twice and redispersed in 800 μL of 10 mM CTAC (23548 × $g$, 30 min). Finally, 20 mL of 100 mM CTAC, 100 μL of 10 nm Au nanospheres, 100 μL of 50 mM NaBr, 1.3 mL of 10 mM AA, and 20 mL of 0.5 mM HAuCl$_4$ were mixed with stirring for 30 min. The resulting solution was washed twice by centrifugation (7168 × $g$, 20 min).

**Synthesis of 1$^{st}$ and 2$^{nd}$ Au@Pt Nanoparticles from Au Octahedral Nanoparticles.** 1$^{st}$ and 2$^{nd}$ Au@Pt nanoparticles were fabricated in a stepwise manner using Au octahedra as seed particles. To synthesize 1$^{st}$ Au@Pt nanoparticles, 2.5 mL of 50 mM CTAB, 1 mL of Au octahedron, 3 μL of 2 mM AgNO$_3$, and 96 μL of 0.1 M AA were mixed in a vial in the presence of iodide ions (50 μM). Every mixing of reagents was conducted through gentle shaking with a vortex shaker. After incubation at 70 °C for 1 h, 96 μL of 0.1 M HCl and 30 μL of 2 mM H$_2$PtCl$_6$ were added to the solution and were incubated for 4 h. After the completion of the reaction, the solution was centrifuged twice at 7168 × $g$ for 10 min and dispersed in DIW while retaining the same solution volume (optical density: 0.8, wavelength: 654 nm). Before the 2$^{nd}$ deposition of Pt, well-faceted overgrowth of Au step was applied to the 1$^{st}$-Au@Pt-O-NPs. Specifically, 1 mL of 0.2 M CTAC, 100 μL of 2 mM AgNO$_3$, 10 μL of 0.1 M AA, and 100 μL of 2 mM HAuCl$_4$ were sequentially added to the previous solution consisting of 1$^{st}$-Au@Pt-O-NPs, and the mixture was kept at 50 °C for 1 h. The solution was centrifuged twice at 7168 × $g$ for 5 min and dispersed in DIW while retaining the same volume (optical density: 1.3, wavelength: 587 nm). Subsequently, Pt was deposited on the resulting nanoparticles. To accomplish this, 2.5 mL of 50 mM CTAB, 8 μL of 2 mM AgNO$_3$, and 96 μL of 0.1 M AA were added to the reaction solution containing the Au nanostructures obtained after well-faceted overgrowth of the Au step. After incubation at 70 °C for 1 h, 96 μL of 0.1 M HCl and 80 μL of 2 mM H$_2$PtCl$_6$ were added and kept at 70 °C for 4 h. After the reaction, the solution was centrifuged twice at 7168 × $g$ for 5 min and dispersed in DIW with constant volume (optical density: 0.8, wavelength: 694 nm).

**Synthesis of 1$^{st}$, 2$^{nd}$, and 3$^{rd}$ Au@Pt nanoparticles from Au truncated octahedral nanoparticles.** To synthesize 1$^{st}$, 2$^{nd}$, and 3$^{rd}$ Au@Pt nanoparticles from Au truncated octahedral nanoparticles, Pt deposition and well-faceted overgrowth were applied to Au truncated octahedral nanoparticles repeatedly. Be aware of following the order row in Supplementary Tables 3–6 for each synthetic step; e.g., to synthesize 2$^{nd}$-Au@Pt-TO-NPs, the reagents in 2$^{nd}$ row in Supplementary Table 3 should be added to the previous step solution. Detailed synthetic steps are described below in an alphabetical order. Every mixing of reagents was conducted through gentle shaking with a vortex shaker.

A. Reagents of 1$^{st}$ row in Supplementary Table 3 were added in order for rim-selective deposition of Pt on Au NPs (1$^{st}$-Au@Pt-TOh-NPs). In detail, after adding AgNO$_3$ and AA, the solution was kept at 70 °C for 1 h. Then, 0.1 M HCl and 1 mM H$_2$PtCl$_6$ were added, and the solution was kept at 70 °C for 4 h. The resulting solution was washed twice by centrifugation (7168 × $g$, 10 min) and redispersed in DIW (optical density and peak absorption wavelength are described in Supplementary Table 3).

B. Reagents of 2$^{nd}$ row in Supplementary Table 4 were sequentially added to 1 mL of the solution from the previous step to synthesize 2$^{nd}$-Au-TO-NPs. The mixed solution was kept at 50 °C for 1 h, and washed twice by centrifugation (7168 × $g$, 10 min). Every solution after well-faceted overgrowth of Au was dispersed in DIW to reach optical density of 1.0.

C. Reagents of 2$^{nd}$ row in Supplementary Table 3 were added to prepare 2$^{nd}$-Au@Pt-TOh-NPs. Synthetic details are the same as A.

D. 3$^{rd}$-Au-O-NPs were prepared by mixing 2$^{nd}$-Au@Pt-TOh-NPs solution with reagents of 3$^{rd}$ row in Supplementary Table 4. Synthetic details are the same as B.

E. Finally, 3$^{rd}$-Au@Pt-O-NPs were synthesized by adding reagents of 3$^{rd}$ row in Supplementary Table 3 to the solution from D. Synthetic details are the same as A. After following the steps mentioned above, the resulting Au@Pt NPs were dispersed in 1 mL of 50 mM CTAB in the presence of 50 μM of iodide ions to prepare the next etching step.

**Synthesis of 1$^{st}$, 2$^{nd}$, 3$^{rd}$, and 4$^{th}$ Au@Pt Nanoparticles from Au Cubic Nanoparticles.** To synthesize 1$^{st}$, 2$^{nd}$, 3$^{rd}$, and 4$^{th}$ Au@Pt nanoparticles from Au cube nanoparticles, Pt deposition and well-faceted overgrowth were performed repeatedly following the same synthetic details of the previous chapter. After selective Pt deposition step (F, H, J, L), follow the optical density and wavelength of Au@Pt NPs listed in Supplementary Table 5. In addition, after well-faceted overgrowth step (G, I, K), the solution was dispersed in DIW with an optical density of 1.0.

A. To deposit Pt on the edges of Au cubes (1$^{st}$-Au@Pt-C-NPs), reagents of 1$^{st}$ row in Supplementary Table 5 were added sequentially to the previously synthesized 1 mL of Au cube solution. AgNO$_3$ and AA were mixed in order and the solution was kept at 70 °C for 1 h. After 1 h, 0.1 M HCl and 1 mM H$_2$PtCl$_6$ were added, and the solution was kept at 70 °C for 4 h. The resulting solution was centrifuged twice (6300 × $g$, 10 min) and dispersed in DIW (optical density and peak absorption wavelength are described in Supplementary Table 5).

B. Then, well-faceted overgrowth of Au was conducted by adding reagents of 2$^{nd}$ row in Supplementary Table 6 into the resulting solution of F to obtain 2$^{nd}$-Au-CO-NPs. The reaction proceeded at 50 °C for 1 h, and washed twice by centrifugation (6300 × $g$, 10 min). Every solution after well-faceted overgrowth of Au was dispersed in DIW to reach an optical density of 1.0.

C. 2$^{nd}$-Au@Pt-CO-NPs were synthesized by selective deposition of Pt. Reagents of 2$^{nd}$ row in Supplementary Table 5 were mixed thoroughly. Same detailed synthetic methods in F were applied in this step.

D. Reagents of 3$^{rd}$ row in Supplementary Table 6 were added to the previous solution. Then, 3$^{rd}$-Au-TO-NPs could be synthesized by following the same synthetic steps of G.

E. 3$^{rd}$-Au@Pt-TO-NPs were obtained by mixing reagents of 3$^{rd}$ row in Supplementary Table 5 with the solution of I), following synthetic procedures of F.

F. Synthesis of 4$^{th}$-Au-O-NPs were accomplished through adding reagents of 4$^{th}$ row in Supplementary Table 6 into the resulting solution of J. Detailed method follows the steps in G.

G. Finally, 4$^{th}$-Au@Pt-O-NPs were obtained by mixing reagents of 4$^{th}$ row in Supplementary Table 5 with the solution from K. Synthetic details follow the procedure in F. After finishing the above steps, the resulting Au@Pt NPs were dispersed in 50 mM CTAB in the presence of 50 μM of iodide ions.

**Synthesis of 1$^{st}$, 2$^{nd}$, 3$^{rd}$, and 4$^{th}$ Pt Nanoframes.** To selectively etch the interior Au domains of Au@Pt nanoparticles, 200 μL of 2 mM HAuCl$_4$ was added to 1 mL of 1$^{st}$, 2$^{nd}$, 3$^{rd}$, and 4$^{th}$ Au@Pt nanoparticles solution with a gentle shaking. The mixed solutions were incubated at 50 °C for at least 1 h. The Pt nanoframes were collected by centrifugation twice and redispersed in 1 mL of 50 mM CTAC for further use.

**Ag deposition of 1$^{st}$, 2$^{nd}$, 3$^{rd}$, and 4$^{th}$ Ag nanoframes.** To deposit Ag, 12 μL of 50 mM NaOH, 120 μL of 2 mM AgNO$_3$, and 120 μL of 10 mM AA were added to the 1 mL of CTAC-dispersed Pt nanoframe solution sequentially. The solution was kept at 30 °C for 1 h and washed with DIW twice after completion of the reaction.

## Electric field simulations

For the electromagnetic simulation, finite element method (FEM) simulation was carried out using commercially available software (COMSOL Multiphysics, Wave Optics Module). Studies were performed in 3D under the wavelength domain. The simulation domain was composed of nanoframe particle and water domain truncated using a perfectly matched layer (PML). In addition, the scattering boundary condition was also applied to avoid undesired scattering effects at the domain boundary. In the simulation, all geometric parameters were obtained from SEM or TEM images. When performing the simulations on the surface charge distribution and electromagnetic field distribution, the nanoframes comprised gold (Au), silver (Ag), and platinum (Pt). The atomic ratio of each component was measured by EDS. The complex refractive index of each material was taken from the data of Johnson and Christy[34] or Rakić[35]. In addition, the refractive index of water was assumed to be 1.33. Meanwhile, for simplicity, only Ag and Pt were chosen as the elements of nanoframes when we calculated the extinction spectra. Based on the TEM images and EDS analysis data, the thickness of the Pt scaffold and Ag layer were assumed to be 10 and 2 nm, respectively. To calculate the extinction spectra of the nanoframes, absorption ($\sigma_{abs}$) and scattering ($\sigma_{sca}$) cross-sections were firstly calculated. (1)$\sigma_{abs}$ and (2)$\sigma_{sca}$ can be defined as

$$\sigma_{abs} = \frac{W_{abs}}{P_{inc}}, \tag{1}$$

$$\sigma_{sca} = \frac{W_{sca}}{P_{inc}} \tag{2}$$

where $P_{inc}, W_{abs}, W_{sca}$ is incident irradiance, energy rate absorbed by the nanoframes, scattered energy rate, respectively. The extinction cross-section (3) $\sigma_{ext}$ can be obtained as the sum of the absorption and scattering cross-section, i.e.,

$$\sigma_{ext} = \sigma_{abs} + \sigma_{sca}. \tag{3}$$

## Single-particle SERS measurement

The single-particle SERS measurement was conducted using a Raman microscope (Ntegra, NT- MDT) equipped with an inverted optical microscope (IX 73, Olympus). A dichroic mirror directs the excitation laser beam into an oil-immersion objective (UPlanSApo, 100×, 1.4 numerical aperture), which focuses the beam to a diffraction-limited spot (~2 μm) on the upper surface of the cover glass slip. Photo-multiplier tube images were obtained using a piezoelectric $x, y$ sample scanner to identify nanoparticles. For attaching Raman reporter on the entire surface of nanostructures, we added the Raman reporter into solution including the resulting nanostructures and kept the solution at 30 °C during 3 h. After centrifugation twice, we can remove the extra amount of Raman reporter in solution and get the concentrated solution of nanostructure. Then, we dropped the solution on the cover glass and removed the droplet using blow gun after 5 min. The SERS spectra were acquired with 633 nm laser (He−Ne laser, Thorlabs) excitation for 10 s. The signals were obtained by a CCD detector (1024 × 256 pixels; Peltier; cooled to −70 °C, Andor Newton DU920P BEX2-DD). After analysis, FESEM images of the samples were obtained after Pt layer deposition using an Ar plasma sputter-coater (Cressington 108 auto) with a current level of 30 mA for 60 s on a slide glass.

## The calculation method for enhancement factor

The enhancement factor of Raman signals was based on the following equation:

$I_{sers}$ and $I_{bulk}$ are the Raman intensities of 2-naphthalenethiol (2-NTT) for surface adsorption and solution samples, respectively.

$N_{bulk}$ is the number of 2-naphthalenethiol molecules in the volume for the normal Raman signals, and $N_{sers}$ is the number of 2-naphthalenethiol molecules absorbed on the Ag nanoframes with different number of nanoframes for the SERS signal. The SERS peak, which is a ring stretching mode at ~1380 cm$^{-1}$, was chosen for the EF calculation.

$N_{bulk}$ was calculated using the following equation:

$$N_{bulk} = \left(\frac{V \times d}{M_w}\right) \times N_A = \left(\frac{\pi \times r^2 \times h \times d}{M_w}\right) \times N_A = 2.67 \times 10^{12} \tag{4}$$

Here, $V$ is the excitation volume of the 2-NTT substrate, $d$ is the density of 2-NTT (1.2 g/cm$^3$), $M_w$ is the molecular weight of 2-NTT (160.23 g/mol), $N_A$ is Avogadro's number (6.02 × 10$^{23}$ mol$^{-1}$, $r$ is the radius of the laser beam (~2 μm), and $h$ is the focal depth of the laser (~47 μm).

$N_{sers}$ was calculated using the following equation:

$$N_{sers} = A \times D \times N_A = 1.3 \times 10^4 \, (\text{1}^{st} \text{ Ag nanoframes}), 1.6 \times 10^4 \, (\text{2}^{nd} \text{ Ag nanoframes}),$$
$$2.9 \times 10^4 \, \left(\text{3}^{rd} \text{ Ag nanoframes}\right), \text{ and } 2.9 \times 10^4 (\text{4}^{th} \text{ Ag nanoframes})$$
$$\tag{5}$$

Here, $A$ is the SERS active area of the nanostructures, which was measured as the entire surface area, 1$^{st}$ Ag nanoframes (0.4 × 10$^{-10}$ cm$^2$), 2$^{nd}$ Ag nanoframes (0.52 × 10$^{-10}$ cm$^2$), 3$^{rd}$ Ag nanoframes (0.93 × 10$^{-10}$ cm$^2$), and 4$^{th}$ Ag nanoframes (0.94 × 10$^{-10}$ cm$^2$). $D$ is the estimated coverage of 2-NTT molecules in a single-assembled monolayer on Ag (5.44 × 10$^{-10}$ mol/cm$^2$)[36].

## Reporting summary

Further information on research design is available in the Nature Research Reporting Summary linked to this article.

## Data availability

The data that support the findings of this study are available from the corresponding author upon request.

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

## Acknowledgements

The authors are grateful to the Nam's lab in Seoul National University for providing a single-particle Raman spectroscope. Sungjae Yoo is grateful to the POSCO Science Fellowship of POSCO TJ Park Foundation, and the Postdoctoral Research Program of Sungkyunkwan University (2022). This research was supported by the National Research Foundation of Korea (NRF) grant funded by the Korean government (MSIT) (No. NRF-2021M3H4A4079145) (S.P.), the National Research Foundation of Korea (NRF) grant funded by the Korea government (MSIT) (NRF-2022R1A2C2002869) (S.P.), the National Research Foundation of Korea (NRF) grant funded by the Korea government (MSIT) (No. NRF-2022R1C1C2003784) (I.J.), and the National Research Foundation of Korea (NRF) grant funded by the Korea government (MSIT) (Grant number: NRF- 2019R1F1A1058851) (J.W.L.).

## Author contributions

S.Y., J.L., and H.H. performed the experiments and data analysis. S.Y. wrote the original draft and S.Y., S.P., I.J., and J.L. reviewed and edited the manuscript. W.P., S.C., and J.W.L. simulated FEM method for electromagnetic field enhancement. S.P. designed and supervised the project.

## Competing interests
