## [Peer Review File · Nature Communications]

REVIEWER COMMENTS

Reviewer #1 (Remarks to the Author):

In this manuscript, Yoo et al. describe the synthesis of complex Pt nanoframes where up to 4 nanoframes can be interconnected into a single nanoobject. The preparation of similar “multi-frame” or “frame-in-frame” (or even nanomatryoshka) structures have been reported before (by the same authors and by others: Chem. Mater. 2014, 26, 12, 3618–3623, small2015, 11, No. 22, 2593–2605, Adv.Mater.2018, 30, 1800939, Nano Research 2016, 9(2): 415–423), but never with such a high degree of control over the morphology and the relative interconnection of the different layers, which certainly represents the most relevant result. Due to the poor plasmonic nature of Pt, the produced frames are finally covered with a Ag layer, which allows for a characterization of the SERS performance of the produced structures.

Overall, the work is certainly worth of consideration by Nature Communications, although I believe it requires some additional work and clarifications before publication.

SYNTHESIS SECTION

- Page 3: “[...] protruded sites can facilitate a galvanic replacement reaction between Pt⁴⁺ ions and the thin Ag layer coated on the entire surface of Au octahedral NPs.” The presence of Ag on the surface of gold lacks context and characterization. From the experimental section we know silver is not added during the synthesis of the original gold nanoparticles. The authors should explain how and when the silver ends up on the surface.

- Similarly, page 4: “It should be noted that preferential deposition of Ag on the Pt sites results from the higher catalytic activity of the Pt surface compared to that of the Au surface.” This statement needs more context or at least a reference.

- Supporting Figure S1-5-7 nicely show the tunability of the gold nanoparticle dimension during the “well-faceted overgrowth of Au” reaction. However, as explain clearly by the author in Fig 2E, we can see how for the smaller particles the overgrowth does not yield complete facets, but instead a thicker gold frame above from the Pt frame generated in the previous step. I wonder if the lack of a complete surface would affect the following Pt deposition. In other words, I would expect that there is a size-limitation to how small the Au-NPs can be made for each generation. If that’s the case it should be discussed.

- Related to the previous point. I wonder if a similar versatility can be introduced in the “rim-selective growth of Pt”. How crucial is the size of the frame? This is also an important point for the reproducibility of the results by other labs.

- The authors underline in multiple occasions the versatility of their system. Looking at Figure 1 it seems that an important limitation to this versatility is the forced evolution of the growth particles from 100-dominated crystals to 111-dominated ones. In other words, it is possible to obtain 4th-Pt-C:CO:TO:O-NF, but not a 4th-Pt-O:TO:CO:C-NF. Do the authors foresee any strategy to overcome this limitation? How generalizable is their approach?

SERS-PERFORMANCE SECTION

- The idea of silver-coating the Pt nanoframes to improve their plasmonic properties is certainly a valuable idea. However, I wonder how much this step could compromise such intricate structures. In particular, I wonder if silver could possibly create bridges between the different concentric frames, especially considering the quite high % of Ag reported in fig S10. I think a tomographic reconstruction videos of the 4th generation structure after silver coating should be added to the revised version.

- Related to the previous point. The near-field simulations reported in Fig 5D-E and S13 are computed for “perfectly oriented” frames. I wonder how much this impact the plasmonics of the frames, as I suspect that experimentally the orientation of the various layers might be slightly misaligned (and maybe made worst during silver coating?).

- The UV-vis shown in Fig5A would be better contextualized if compared to simulated extinction of the various structures.

- Looking at Figure S11, nanoparticles seems to be quite close to one another, considering a 2 um spot size. What is the estimated spatial resolution of the Raman setup? The claim of single-particle SERS should be better supported by the authors.

- The authors claims highly uniform and reproducible SERS signals from Figure S12, although quite significant particle-to-particle variation can be clearly observed. Authors should try to quantify the reproducibility of their SERS signals as standard deviation of the most intense peak.

- Authors should comment on the choice of a 633 laser for their characterization.

EXPERIMENTAL SECTION

Several experimental details are missing, including the precise order of addition of the reagents, the stirring conditions, and the specification of how exact temperature is maintained during growth.

In general, I think the experimental section should be revised thoroughly. As it is, I think it would be quite hard to reproduce the reported protocols (if not impossible for not skilled hands). In particular, a UV-vis characterization of the various step should be reported (without normalization), as it would help in the reproducibility and in checking the quality of the colloids at each step, as well as (even more importantly) to estimate the concentration of the various colloids after centrifugation and cleaning steps. I suggest the author to take ACS Nano 2021, 15, 12, 18600–18607 as a model for their experimental section.

Some example of missing information are:

1) Au Octahedra, truncated octahedra, cubes:

- please, specify which solvent is used to dilute the seeds 100 times. Is it Milli-Q water? Or 75mM CTAB solution? Why seeds are not diluted for the cubes?

- Starting gold nanoparticles. Each synthesis is reported for a different volume of final product. It would be much easier to compare growth conditions if all reagents amount are reported for a same volume of product, unless there is a specific reason that prevent to synthesize a certain volume of a particular starting gold nanoparticles, in which case this should be reported.

- Between the first and the second growth step, are the particle cleaned or concentrated in any way? For example, this is the case for the cubes. Centrifugation steps are knowingly hard to reproduce, and can lead to drastically different concentration of particles for the following steps. Please, report a UV-vis characterization of all the synthetic steps, in order for the synthesis to be properly reproduced.

2) Synthesis of Nth-nanoframes.

- the authors report an optical density of 1.2 for Au octahedron (similar data should be provided for truncated octahedral and cubes). Moreover, please, specify at what wavelength the optical density is taken. Once again, a complete UV-vis characterization would be very helpful to ensure reproducibility

- 1st Au@Pt nanoparticles are synthesized in CTAB, but the following Au deposition is performed by adding Au@Pt nanoparticles "dispersed in CTAC". Authors should explain how this change in surfactant is performed, and how the concentration of the particles in the final solution used for the next synthetic step is estimated (a very similar comment can be made after the gold deposition, as the following Pt growth is once again performed in CTAB).

MINOR COMMENTS:

- I would suggest the author to change the Title, as I find it very much not informative on the content of the manuscript.

- Also, I am not completely sold on the "4-dimensional" definition of the produced particles, as it is not clear to me which one is the 4th dimension. I would try to better explain the choice of 4D and Nth-nanoframes in the text.

- In the abstract and the introduction there is an extensive use of quite vague terms and statements (various kind, selective deposition of metals atoms, inner metals). I would invite the authors to be more specific.

Despite my lengthy critique of the manuscript, I really think the research presented here is of excellent value, and scientifically sound. I look forward to see this published in Nature communications after revisions.

Reviewer #2 (Remarks to the Author):

The manuscript 'N-th Nanoframes' reports on a novel synthetic strategy to build four-dimensional complex nanoframes with high homogeneity in shape and size. Their approach enables complex nanoframes to be built from a single type of Au solid nanostructures (such as octahedra, truncated octahedra, cuboctahedra and cubes) to nanoframes that consist of four different geometrical morphologies, making this strategy highly customizable.

The synthesis of complex nanoframes with multiple geometric shapes while also possessing high structural controllability is rare and challenging. However, this manuscript has achieved unprecedented four-dimensional complex nanoframes with excellent control over the structure and size. In addition, using 2-naphthalenethiol as an analyte for single-particle surface-enhanced Raman scattering (spSERS), the authors have clearly demonstrated the high electromagnetic near-field enhancement within their 4D complex Ag nanoframes and the potential of nanoframes for future applications. All in all, the experiments and discussions are well-written and insightful. The manuscript is acceptable for publication with the following issues to be addressed.

1. While investigating the well-faceted overgrowth of Au such as the results presented in Supplementary Fig. S2 and S4, the authors experimented with three different growth patterns – CTAC with and without Ag ions but CTAB with Ag ions only. What about the experiment on the growth pattern involving CTAB without Ag ions? What would be the expected results and differences?
2. After the comproportionation reaction of Au⁺ ions, what is the typical percentage of Au that would be etched and the percentage of Au that would be residual? Would the residual Au affect the performance of the nanoframes or its future applications?
3. In Supplementary Fig. S6 which discusses the control of intra-nanogap distance through tailoring the degree of well-faceted overgrowth of Au, are there any studies done on the effect of varying the intra-nanogap distance of the same type of nanoframe on their enhancement of electromagnetic near-field?
4. For 4D complex Ag nanoframes with different number of inner nanoframes in a single entity, the authors mentioned that the typical Ag atomic fractions of the 1st, 2nd, 3rd and 4th of Ag nanoframes were 35%, 30%, 42% and 30%. Are there any reasons why certain orders of the nanoframes have a higher percentage of Ag deposited on their Pt scaffolds?

5. In addition, for the 4D complex nanoframes, does 4th-Pt-C:CO:TO:O-NFs possess the optimal shapes from the core to the outer shape? How is this order of the four different nanostructures from the core to the outermost shape determined?

6. In Fig. 5C which illustrates the data of single-particle Raman intensity at 1379 cm^{-1} as a function of the number of nanoframes, why is the error bar for the Raman intensity so huge when 3 and 4 nanoframes are used?

7. What is the SERS enhancement factor of the nanoframes for single-particle SERS detection of 2-naphthalenethiol?

8. Based on the current approach, it seems like there could be an infinite number of nanoframes embedded in a single entity. Since the manuscript only presents nanoframe up to a maximum of 4th order, is four the maximum number of nanoframes that can be embedded in a single entity? If so, what is the limiting factor involved here? Otherwise, what is the maximum number of nanoframes that can be achieved in a single entity using such an approach?

9. Since this approach can be expanded to nanostructures with different geometric shapes, what do they think would be different if another material such as Ag solid nanostructures instead of Au are used as starting materials instead?

Reviewer #3 (Remarks to the Author):

In this manuscript, the authors reported the synthesis of three-dimensional polyhedral nanoframes embedded with different numbers and shapes of nanoframes, which are named "N-th nanoframes" by the authors. By tailoring the growth behavior of Pt(0) and Au(0) atoms through kinetic control, the authors managed to obtain "N-th nanoframes" with different parameters. These "N-th nanoframes" were then coated with Ag and evaluated as a single-particle platform for SERS detection. The result shows that the intensity of Raman signals of 2-naphthalenethiol would be marked amplified as the number of inner nanoframes increases. The rational design and fabrication of complex nanoframes and their enhanced Raman signal for SERS detection is quite interesting. Generally, I would like to recommend its publication in Nat. Commun..

Selective deposition of noble metals on preformed seeds has been well documented in many reported works. The originality of the present work lies in the delicate design of complex nanoframe structures and their superior performance in SERS. Although the synthetic protocol and the complex nanoframe structure have been well characterized and organized by the authors, the structure-related SERS performance, in my opinion, is not fully discussed in the current version. A number of critical flaws are detailed below.

- 1) The intensity of the SERS signal depends on the number of adsorbed molecules. The surface area of "N-th nanoframes" increases with the number "N", resulting in the different number of adsorbed molecules during SERS signal detection. Enhancement factors of different "N-th nanoframes" should be calculated to reveal the different capabilities of complex nanoframes towards SERS. This is the main flaw in this study.
- 2) As this work mainly focuses on the rational design of excellent SERS substrates. Here, it is better to compare the SERS enhancement factor with other previously reported materials, such as Ag nanomaterials.
- 3) Another concern is about the composition of the nanoframes. The composition of the nanoframes play an important role in SERS. Here, EDS was introduced to determine the atomic ratio of Au, Ag, and Pt in nanoframes. As we know, EDS is useful in analyzing the composition of nanostructured materials, however, do not have sufficient accuracy. Other characterizations, such as ICP-MS, are required to reveal the atomic ratios of different elements in "N-th nanoframes".

Collectively, this work provides a potential route for the construction of complex nanostructures to obtain highly-sensitive SERS materials for chemical detection. However, the weakness in the SERS part strongly limited the importance of the work, and the matters mentioned above concern the key idea and conclusions of the paper; hence it is recommended for publication after a major revision.

Reviewer #4 (Remarks to the Author):

In the manuscript "N-th nanoframe", Sungjae Yoo, Jaewon Lee, Hajir Hilal, Insub Jung, Woongkyu Park, Joong Wook Lee, Soobong Choi, and Sungho Park describe the synthesis of multi-walled nanocages from polyhedra. Despite a small lack of details, the synthesis is elaborately performed and matches state of the art colloidal synthesis and finds itself in the field of physical chemistry and plasmonics well.

However, as already obvious by the too minimalistic title, novelty, uniqueness, and a groundbreaking application are missing and I don't see much advantages as compared to other nanorattles, nanocages, and especially nanomatryoshkas, which are well known in literature. Exploiting Pt frames as bars and the

selective deposition of silver seems like a novel and smart way of fabrication, but I think it is too specialized for the broad readership of Nature Communications, but rather would fit in a journal focusing on physical chemistry.

Independently, the following points should be addressed, clarified, and discussed:

1. In what physical sense are these nanocages 4 dimensional?

2. The thin Ag layer enabling faceted and epitaxial growth needs to be proven by HR-TEM directly showing this layer and alternative mechanisms, like UPD, need to be ruled out.

For the presented EDX mappings (e.g. Fig S3 and S10) the signal-to-noise ratio is too low in order to proof a homogeneous and epitaxial growth. Further, the EDX spectra should be shown in the supporting information, especially since the signals of Pt(M) and Au(M) are close, as well as those of Ag(L) and Cl(K), and thus, do not allow clear separation in all cases.

3. Is the inner frame fixed in position or is it freely movable in the outer frame? (e.g. Fig S6 looks collapsed)

If it is fixed, how is it fixed and how do you guarantee stable fixation? How does this fixation influence the plasmonics, as most probably it will happen on the smallest distance between the rims?

If it is free, how is a collapse of the well-defined structure and plasmonic hotspots avoided and ensured that the rims are separated during the final silver overgrowth?

4. How thick and homogeneous is the silver shell for the SERS measurements? HR-TEM (and EDX) are inevitable, also to exclude roughness as source of field enhancement.

5. The optical properties of all particle stages should be discussed and shown, especially in regard to the highly damping Pt. How thick, does the silver shell around those bars need to be, in order to dominate the plasmonics?

6. As plasmonics are the main application of these particles, an in-depth discussion of the resulting plasmonic modes should be included and a (hybridization) model explaining their nature should be proposed.

7. As the laser for the SERS measurements matches best to higher order frames, how is excluded, that this is a main source of the increased intensity?

REVIEWER COMMENTS

Reviewer #1 (Remarks to the Author):

In this manuscript, Yoo et al. describe the synthesis of complex Pt nanoframes where up to 4 nanoframes can be interconnected into a single nanoobject. The preparation of similar “multi-frame” or “frame-in-frame” (or even nanomatyoshka) structures have been reported before (by the same authors and by others: Chem. Mater. 2014, 26, 12, 3618–3623, small2015, 11, No. 22, 2593–2605, Adv.Mater.2018, 30, 1800939, Nano Research 2016, 9(2): 415–423), but never with such a high degree of control over the morphology and the relative interconnection of the different layers, which certainly represents the most relevant result. Due to the poor plasmonic nature of Pt, the produced frames are finally covered with a Ag layer, which allows for a characterization of the SERS performance of the produced structures.

Overall, the work is certainly worth of consideration by Nature Communications, although I believe it requires some additional work and clarifications before publication.

SYNTHESIS SECTION

1. Page 3: “[...] protruded sites can facilitate a galvanic replacement reaction between Pt⁴⁺ ions and the thin Ag layer coated on the entire surface of Au octahedral NPs.” The presence of Ag on the surface of gold lacks context and characterization. From the experimental section we know silver is not added during the synthesis of the original gold nanoparticles. The authors should explain how and when the silver ends up on the surface.

Answer: As following the reviewer’s comment, we changed the sentence to explain the synthetic mechanism of rim-selective growth of Pt in detail. It says “In the first “rim-selective growth of Pt” step, Ag⁺ ions were added to the reaction solution in the presence of ascorbic acid resulting in the formation of thin Ag layer on the whole surface area of Au NPs. Subsequently, Pt atoms were selectively deposited along the edges and vertexes of Au octahedral NPs because the relatively higher surface energy of those protruded sites can facilitate a galvanic replacement reaction between Pt⁴⁺ ions and the thin Ag layer, leading to 1st-Au@Pt-O-NPs with a total size of 77 ± 2 nm (Fig. 2B).” The sentence is highlighted with red color in the manuscript (Page 3)

2. Similarly, page 4: “It should be noted that preferential deposition of Ag on the Pt sites results from the higher catalytic activity of the Pt surface compared to that of the Au surface.” This statement needs more context or at least a reference.

Answer: As following reviewer’s comment, we added the references (J. Phys. Chem. C, 2016, 120, 41, 23698–23706, Applied Surface Science, 229, (2004), 34–42) which show the correlation between surface energy and surface chemical reactivity.

3. Supporting Figure S1-5-7 nicely show the tunability of the gold nanoparticle dimension during the “well-faceted overgrowth of Au” reaction. However, as explain clearly by the author in Fig 2E, we can see how for the smaller particles the overgrowth does not yield complete facets, but instead a thicker gold frame above from the Pt frame generated in the previous step. I wonder if the lack of a complete surface would affect the following Pt deposition. In other words, I would expect that there is a size-limitation to how small the Au-NPs can be made for each generation. If that’s the case it should be discussed.

Answer: As reviewer mentioned, before forming the well-defined terrace regions on the Au@Pt nanoparticles, Au atoms are selectively deposited along the Pt sites of Au@Pt nanoparticles generating the thicker Au nanoframes as depicted in Fig 2E. Through controlling the amount of Au precursor in the reaction solution, the degree of Au overgrowth could be tailored as shown in below Figure B to D. When Pt atoms are deposited on the intermediated nanostructures corresponding to Figure B, Pt atoms are selectively grown not only on the outer edge sites but also on inner edge sites of the thicker Au nanoframes as shown in Figure E. Subsequently, after etching inner Au, another complex nanoframes could be observed as shown Figure F. Recently, we are conducting the research about these unexpected nanoframe structures.

4. Related to the previous point. I wonder if a similar versatility can be introduced in the “rim-selective growth of Pt”. How crucial is the size of the frame? This is also an important point for the reproducibility of the results by other labs.

Answer: To investigate the effect of Pt thickness on Au overgrowth pattern, Pt thickness of Au@Pt nanoparticles were tuned from 13 nm to 25 nm as show in Figure A to C followed by overgrowth of Au, forming the well-faceted Au nanoparticles (Figure D, E, and F). Taken together, it demonstrates that effects of Pt thickness on Au overgrowth pattern is most negligible. But, when we use Pt thickness with below 13 nm, the resulting nanoframe structures could not retain their framework after “etching inner Au” step, resulting in occurring the debris shown in Figure G.

5. The authors underline in multiple occasions the versatility of their system. Looking at Figure 1 it seems that an important limitation to this versatility is the forced evolution of the growth particles from 100-dominated crystals to 111-dominated ones. In other words, it is possible to obtain 4th-Pt-C:CO:TO:O-NF, but not a 4th-Pt-O:TO:CO:C-NF. Do the authors foresee any strategy to overcome this limitation? How generalizable is their approach?

Answer: Thank you for your good question. The reference (J. AM. CHEM. SOC. 2008, 130, 6949–6951) show that when Pd atoms are deposited on the Au octahedron, octahedron shape could be transformed into truncated cube then into cube. Based on this reference, it can be expected that if this synthetic reaction considers as a new chemical toolkit and combine with our multiple synthetic pathways, such 3rd-Pt-O:TO:O-NF or 4th-Pt-O:C:CO:TO-NF would be realized. Furthermore, if anisotropic nanoparticles utilize as a start material, structural controllability of complex nanoframe structures would be extended further and it will demonstrate the generality of our synthetic approach.

SERS-PERFORMANCE SECTION

6. The idea of silver-coating the Pt nanoframes to improve their plasmonic properties is certainly a valuable idea. However, I wonder how much this step could compromise such intricate structures. In particular, I wonder if silver could possibly create bridges between the different concentric frames, especially considering the quite high % of Ag reported in fig S10. I think a tomographic reconstruction videos of the 4th generation structure after silver coating should be added to the revised version.

Answer: As following reviewer's comment, we obtained the tomographic reconstruction video for 4th Ag nanoframes and added to Supplementary Video 4. It shows that four different nanoframes are arranged in single entity and connected by metal bridges. Additionally, through EDS image mapping data, it could be confirmed that Ag atoms were coated the whole surface of nanoframes as well as bridges among the nanoframes.

7. Related to the previous point. The near-field simulations reported in Fig 5D-E and S13 are computed for “perfectly oriented” frames. I wonder how much this impact the plasmonics of the frames, as I suspect that experimentally the orientation of the various layers might be slightly misaligned (and maybe made worst during silver coating?).

Answer: We also think that there are unexpected structural variations such as slightly misaligned orientation of nanoframes resulting in variations in single-particle SERS signal as shown in Supplementary Fig.18. However, this does not significant impact the trends of SERS intensity as regard to number of nanoframes.

8. The UV-vis shown in Fig5A would be better contextualized if compared to simulated extinction of the various structures.

Answer: We obtained the theoretical calculated UV-vis-NIR spectrum for Ag nanoframes with different number of nanoframes through FEM method as shown in below Figure. We discussed and added the sentence saying “Dipole mode of 1st Ag nanoframes blue-shifted from 936 nm to 534 nm resulting from surface plasmon coupling between inner and outer nanoframe. Then, as the number of nanoframes increased from double to quadruple, dipole mode of nanoframes red-shifted from 534 nm to 564 nm and 620 nm accompanying with peak broadening. This trend well agreed with theoretical calculated UV-vis-NIR spectrum obtained by theoretical simulations based on a finite-element method (FEM) as shown in Supplementary Fig. 15. Charge distribution shown in Figure 5 (red: electron rich, blue: electron deficient) clearly indicates that the surface charge density significantly increased in a single entity as the number of nanoframes increased from single to quadruple, which is a distinctive characteristic of N-th nanoframes. We expect that surface plasmon coupling would occur in a complex manner, representing a broad LSPR feature (Fig. 5A).” to manuscript (page 7)

9. Looking at Figure S11, nanoparticles seem to be quite close to one another, considering a 2 μm spot size. What is the estimated spatial resolution of the Raman setup? The claim of single-particle SERS should be better supported by the authors

Answer: Laser beam radius is \sim about 2 μm . We make sure that Raman signal comes from single particle because when laser spot is a slightly deviate from dark spot marked with white circle dot in Supplementary Fig. 16 where particle exist, Raman signal could not appear and thus it demonstrates the nanoparticles around dark spot does not affect the Raman signal.

10. The authors claims highly uniform and reproducible SERS signals from Figure S12, although quite significant particle-to-particle variation can be clearly observed. Authors should try to quantify the reproducibility of their SERS signals as standard deviation of the most intense peak.

Answer: As following the reviewer's comment, we quantified the SERS signals (at 1379 cm^{-1}) as shown in below first Figures. It shows that average and standard deviation of SERS signal for Ag nanoframes with different number of nanoframes is 23 ± 30 (for 1st Ag nanoframes), 157 ± 57 (for 2nd Ag nanoframes), 510 ± 366 (for 3rd Ag nanoframes), and 1571 ± 360 (for 4th Ag nanoframes), respectively. Additionally, enhancement factors were also calculated as shown in second Figures. Among them, typically, 4th Ag nanoframes have the highest enhancement factor (1.3×10^9) and show the reproducibility within two orders of magnitude. We added enhancement factor to Supplementary Fig. 20 and the sentence, saying "Calculated enhancement factors (at 633 nm) are 6.3×10^7 (for 1st Ag nanoframes), 3.7×10^8 (for 2nd Ag nanoframes), 7.6×10^8 (for 3rd Ag nanoframes), and 1.3×10^9 (for 4th Ag nanoframes) (Supplementary Fig. 20)." to manuscript (page 8).

Average and standard deviation of SERS signals

Enhancement factors

11. Authors should comment on the choice of a 633 laser for their characterization.

Answer: We measured single-particle SERS signals of Ag nanoframes with different number of frames at 532 and 785 nm excitation wavelength, respectively. At the 532 nm excitation wavelength, Raman signals increases as the number of frames increases, as shown in first figures. However, at the 785nm excitation wavelength, Raman signals did not appear as shown in second figures. Taken together, it demonstrates that the most well-resolved Raman signals could be observed at 633 nm excitation wavelength as shown Supplementary Fig. 18. We added data showing reproducibility of SERS signals at 532 and 785 nm excitation wavelength to Supplementary Fig. 17 and 19, respectively and changed the sentence from "In addition, SERS signals are obtained by measuring 40 single particles, indicating highly uniform and reproducible SERS signals (Supplementary Fig. 16)." to "SERS signals were obtained at different excitation wavelength, indicating that the highest intensity of Raman signals with relatively high reproducibility SERS signals was observed at the 633 nm excitation wavelength (Supplementary Fig. 17, 18, and 19) in manuscript (page 8). The sentence, saying "Calculated enhancement factors (at 633 nm) are 6.3×10^7 (for 1st Ag nanoframes), 3.7×10^8 (for 2nd Ag nanoframes), 7.6×10^8 (for 3rd Ag nanoframes), and 1.3×10^9 (for 4th Ag nanoframes) (Supplementary Fig. 20)." was added to manuscript (page 8). Calculation methods for enhancement factor are added to experiment sections in manuscript (page 13).

At 532 nm laser excitation,

At 785 nm laser excitation

EXPERIMENTAL SECTION

Several experimental details are missing, including the precise order of addition of the reagents, the stirring conditions, and the specification of how exact temperature is maintained during growth.

In general, I think the experimental section should be revised thoroughly. As it is, I think it would be quite hard to reproduce the reported protocols (if not impossible for not skilled hands). In particular, a UV-vis characterization of the various step should be reported (without normalization), as it would help in the reproducibility and in checking the quality of the colloids at each step, as well as (even more importantly) to estimate the concentration of the various colloids after centrifugation and cleaning steps. I suggest the author to take ACS Nano 2021, 15, 12, 18600–18607 as a model for their experimental section.

Some example of missing information are:

1) Au Octahedra, truncated octahedra, cubes:

- please, specify which solvent is used to dilute the seeds 100 times. Is it Milli-Q water? Or 75mM CTAB solution? Why seeds are not diluted for the cubes?

Answer: Milli-Q water was used to dilute the seed solution. In the case of cubes, dilution step of seed is not necessary according to the reference.

- Starting gold nanoparticles. Each synthesis is reported for a different volume of final product. It would be much easier to compare growth conditions if all reagents amount are reported for a same volume of product, unless there is a specific reason that prevent to synthesize a certain volume of a particular starting gold nanoparticles, in which case this should be reported.

Answer: As following the reviewer's comment, we corrected all reagents amount for a same volume of product.

- Between the first and the second growth step, are the particle cleaned or concentrated in any way? For example, this is the case for the cubes. Centrifugation steps are knowingly hard to reproduce, and can lead to drastically different concentration of particles for the following steps. Please, report a UV-vis characterization of all the synthetic steps, in order for the synthesis to be properly reproduced.

Answer: We added UV-vis characterization (optical density and wavelength) of all synthetic steps in Methods and Supplementary Tables 1 – 4 as following the reviewer's comment.

2) Synthesis of Nth-nanoframes.

- the authors report an optical density of 1.2 for Au octahedron (similar data should be provided for truncated octahedral an cubes). Moreover, please, specify at what wavelength the optical density is taken. Once again, a complete UV-vis characterization would be very helpful to ensure reproducibility

Answer: Wavelength and optical density is 564 nm and 1.2 for octahedral nanoparticles, 539 nm and 1.0 for truncated octahedral nanoparticles, and 546 nm and 1.2 for cubic nanoparticles, respectively. We added these UV-vis characterizations in Methods as following the reviewer's comment to ensure reproducibility.

- 1st Au@Pt nanoparticles are synthesized in CTAB, but the following Au deposition is performed by adding Au@Pt nanoparticles "dispersed in CTAC". Authors should explain how this change in surfactant is performed, and how the concentration of the particles in the final solution used for the next synthetic step is estimated (a very similar comment can be made after the gold deposition, as the following Pt growth is once again performed in CTAB).

Answer: We thoroughly revised **Methods** as the following description because our previous expression could occur a misunderstanding to readers. After Pt deposition or well-faceted overgrowth of Au step, the resulting nanoparticles were washed twice using DIW through centrifugation. The resulting solution was dispersed in DIW again. Then, CTAC or CTAB solution were added to the DIW-dispersed nanoparticle solution to change the surfactant environment after Pt deposition or well-faceted overgrowth of Au, respectively.

MINOR COMMENTS:

- I would suggest the author to change the Title, as I find it very much not informative on the content of the manuscript.

Answer: As following the reviewer's comment, we changed the title from "N-th Nanoframes" to "N-th Nanoframes: Integrating the Polyhedral Multiple Nanoframes into a Single Entity for Amplifying the Electromagnetic Near Field".

- Also, I am not completely sold on the "4-dimensional" definition of the produced particles, as it is not clear to me which one is the 4th dimension. I would try to better explain the choice of 4D and Nth-nanoframes in the text.

Answer: As following the reviewer's comment, to clarify the definition of the resulting complex nanoframes, we removed the word "4-dimensional" in the text and used only "N-th nanoframes" to emphasize structural controllability, typically, the number of nanoframe in a single entity.

- In the abstract and the introduction there is an extensive use of quite vague terms and statements (various kind, selective deposition of metals atoms, inner metals). I would invite the authors to be more specific.

Answer: As following the reviewers' comment, we removed ambiguous terms and changed to "various shapes", "selective deposition of platinum atoms", and "inner Au domains", respectively. The sentence is highlighted with red color in the manuscript (Page 1).

Despite my lengthy critique of the manuscript, I really think the research presented here is of excellent value, and scientifically sound. I look forward to see this published in Nature communications after revisions.

Reviewer #2 (Remarks to the Author):

The manuscript 'N-th Nanoframes' reports on a novel synthetic strategy to build four-dimensional complex nanoframes with high homogeneity in shape and size. Their approach enables complex nanoframes to be built from a single type of Au solid nanostructures (such as octahedra, truncated octahedra, cuboctahedra and cubes) to nanoframes that consist of four different geometrical morphologies, making this strategy highly customizable.

The synthesis of complex nanoframes with multiple geometric shapes while also possessing high structural controllability is rare and challenging. However, this manuscript has achieved unprecedented four-dimensional complex nanoframes with excellent control over the structure and size. In addition, using 2-naphthalenethiol as an analyte for single-particle surface-enhanced Raman scattering (spSERS), the authors have clearly demonstrated the high electromagnetic near-field enhancement within their 4D complex Ag nanoframes and the potential of nanoframes for future applications. All in all, the experiments and discussions are well-written and insightful. The manuscript is acceptable for publication with the following issues to be addressed.

1. While investigating the well-faceted overgrowth of Au such as the results presented in Supplementary Fig. S2 and S4, the authors experimented with three different growth patterns – CTAC with and without Ag ions but CTAB with Ag ions only. What about the experiment on the growth pattern involving CTAB without Ag ions? What would be the expected results and differences?

Answer: As following the reviewer's comment, we conducted the experiment. When the Au ions are reduced on the Au@Pt nanoparticles with different shapes in the presence of CTAB without Ag ions, surface of the resulting nanoparticles could not be faceted because large lattice mismatch between Au and Pt causes the non-epitaxial growth pattern of Au, as shown in below SEM images.

2. After the comproportionation reaction of Au⁺ ions, what is the typical percentage of Au that would be etched and the percentage of Au that would be residual? Would the residual Au affect the performance of the nanoframes or its future applications?

Answer: Typically, percentage of residual Au is found to be ~39 % through TEM EDS analysis. We think the residual would play an important role in photocatalysis applications as a hot electron generator in visible regions. Specifically, Pt which is catalytic metal and Au which is plasmonic metal comprises of

our bimetallic nanostructures. Therefore, the synergistic effect of those two components may raise a promising improvement in a catalyst application.

3. In Supplementary Fig. S6 which discusses the control of intra-nanogap distance through tailoring the degree of well-faceted overgrowth of Au, are there any studies done on the effect of varying the intra-nanogap distance of the same type of nanoframe on their enhancement of electromagnetic near-field?

Answer: The reference (J. Am. Chem. Soc. 2020, 142, 28, 12341–12348) shows the effect of varying the intra-nanogap distance between the Ag inner and outer ring in a Ag double nanoring. Electromagnetic near-field enhances as the intra-nanogap distance decreases, which was proved by a SERS signal enhancement.

We also have confirmed the enhancement of electromagnetic near-field of 2nd-Au-TOh:Oh-NFs having different intra-nanogap distance through single-particle SERS measurement. Specifically, intra-nanogap distance of 2nd-Au-TOh:Oh-NFs decreases as the amount of Au coating the core Pt increases as shown in below SEM images and the corresponding single-particle SERS signal intensity increases as distance of intra-nanogap decreases. Taken together, it demonstrates that enhancement of electromagnetic near-field could be tuned as a function of intra-nanogap distance.

4. For 4D complex Ag nanoframes with different number of inner nanoframes in a single entity, the authors mentioned that the typical Ag atomic fractions of the 1st, 2nd, 3rd and 4th of Ag nanoframes were 35%, 30%, 42% and 30%. Are there any reasons why certain orders of the nanoframes have a higher percentage of Ag deposited on their Pt scaffolds?

Answer: Those percentage of Ag are representative values measured by TEM EDS analysis. Although we tried to adjust the Ag atomic fraction of each sample to have similar values, there are variation in Ag atomic percentage among nanoparticles in each sample.

5. In addition, for the 4D complex nanoframes, does 4th-Pt-C:CO:TO:O-NFs possess the optimal shapes from the core to the outer shape? How is this order of the four different nanostructures from the core to the outermost shape determined?

Answer: Supplementary Video 3 clearly show the shape of each nanoframe for 4th-Pt-C:CO:TO:O-NFs. Shape transformation of Au nanoparticles after well-faceted overgrowth of Au step direct the order of four different nanoframe shape. As shown in Fig. 4, as the multi-stepwise chemical reaction proceed, shape of Au nanoparticle was transformed starting from cube to cuboctahedron, truncated octahedron, and octahedron and utilized as a template for selective growth of Pt resulting in 4th-Pt-C:CO:TO:O-NFs. The order was determined arbitrarily following the synthetic and characterization excellency.

6. In Fig. 5C which illustrates the data of single-particle Raman intensity at 1379 cm⁻¹ as a function of the number of nanoframes, why is the error bar for the Raman intensity so huge when 3 and 4 nanoframes are used?

Answer: This is because polydispersity of nanoframes is increased as the number of synthetic steps increases.

7. What is the SERS enhancement factor of the nanoframes for single-particle SERS detection of 2-naphthalenethiol?

Answer: As following reviewer's comment, we plotted SERS enhancement data for 1st, 2nd, 3rd, and 4th Ag nanoframes as shown in below Figures and added to Supplementary Fig. S20. We added the sentence, saying "Calculated enhancement factors (at 633 nm) are 6.3×10^7 (for 1st Ag nanoframes), 3.7×10^8 (for 2nd Ag nanoframes), 7.6×10^8 (for 3rd Ag nanoframes), and 1.3×10^9 (for 4th Ag nanoframes) (Supplementary Fig. 20)." to manuscript (page 8).

8. Based on the current approach, it seems like there could be an infinite number of nanoframes embedded in a single entity. Since the manuscript only presents nanoframe up to a maximum of 4th order, is four the maximum number of nanoframes that can be embedded in a single entity? If so, what is the limiting factor involved here? Otherwise, what is the maximum number of nanoframes that can be achieved in a single entity using such an approach?

Answer: Currently, the maximum number of nanoframe that can be embedded in a single entity is four. Limiting factor for obtaining quinary nanoframe is the size of nanoparticle. After depositing Au atoms on the 4th Au@Pt nanoparticles corresponding to Figure A, the obtained Au nanoparticles could not retain the well-defined facets (Figure B) and thus Pt atoms were deposited both on edge and terrace regions (Figure C), resulting in shell-like hollow nanostructures after inner etching Au step (Figure D).

9. Since this approach can be expanded to nanostructures with different geometric shapes, what do they think would be different if another material such as Ag solid nanostructures instead of Au are used as starting materials instead?

Answer: Based on this reference (J. AM. CHEM. SOC. 2008, 130, 6949–6951), if Ag are deposited on the 1st-Au@Pt-O-NPs instead of Au in well-faceted overgrowth step, outer shape would be transformed into cube not the octahedron. Therefore, it will be possible to synthesize the different geometric of nanoframes, for example, 2nd-Pt-O:C-NFs.

Reviewer #3 (Remarks to the Author):

In this manuscript, the authors reported the synthesis of three-dimensional polyhedral nanoframes embedded with different numbers and shapes of nanoframes, which are named “N-th nanoframes” by the authors. By tailoring the growth behavior of Pt(0) and Au(0) atoms through kinetic control, the authors managed to obtain “N-th nanoframes” with different parameters. These “N-th nanoframes” were then coated with Ag and evaluated as a single-particle platform for SERS detection. The result shows that the intensity of Raman signals of 2-naphthalenethiol would be marked amplified as the number of inner nanoframes increases. The rational design and fabrication of complex nanoframes and their enhanced Raman signal for SERS detection is quite interesting. Generally, I would like to recommend its publication in Nat. Commun.

Selective deposition of noble metals on preformed seeds has been well documented in many reported works. The originality of the present work lies in the delicate design of complex nanoframe structures and their superior performance in SERS. Although the synthetic protocol and the complex nanoframe structure have been well characterized and organized by the authors, the structure-related SERS performance, in my opinion, is not fully discussed in the current version. A number of critical flaws are detailed below.

1) The intensity of the SERS signal depends on the number of adsorbed molecules. The surface area of “N-th nanoframes” increases with the number “N”, resulting in the different number of adsorbed molecules during SERS signal detection. Enhancement factors of different “N-th nanoframes” should be calculated to reveal the different capabilities of complex nanoframes towards SERS. This is the main flaw in this study.

Answer: As following reviewer’s comment, we plotted SERS enhancement data for 1st, 2nd, 3rd, and 4th Ag nanoframes as shown in below Figures and added to Supplementary Fig. 20. We added the sentence, saying “Calculated enhancement factors (at 633 nm) are 6.3×10^7 (for 1st Ag nanoframes), 3.7×10^8 (for 2nd Ag nanoframes), 7.6×10^8 (for 3rd Ag nanoframes), and 1.3×10^9 (for 4th Ag nanoframes) (Supplementary Fig. 20).” to manuscript (page 8).

2) As this work mainly focuses on the rational design of excellent SERS substrates. Here, it is better to compare the SERS enhancement factor with other previously reported materials, such as Ag nanomaterials.

Answer: As following reviewer's data, we compared SERS enhancement of N-th nanoframe with previously reported other nanoparticles as shown below Figure and we added this data to Supplementary Fig. 21. We added the sentence saying "Among these, the enhancement factor of 4th Ag nanoframes is belonging to the highest ranking compared to that of previously plasmonic materials" to manuscript (page 8).

Type of nanoparticles	Raman analytes	Excitation wavelength	Enhancement factor (EF)	Reference
Au Ag alloy nanourchins	Crystal violet	633nm	$\sim 10^9$	37
Au-Ag core-shell nanodumbbells	Cy3 dye	514 nm	$\sim 10^{13}$	38
Anisotropic Ag nanoparticles	Benzenethiol	514, 633, 785 nm	$3 \times 10^4, 5 \times 10^4$	39
Ag double nanorings	2-Naphthalenthio	785 nm	$1.4 \times 10^8, 2.6 \times 10^8, 5.1 \times 10^8$	29
Ag nanoplates with ultranarrow gaps	2-Naphthalenthio	785 nm	4.1×10^{10}	40
Porous AuAg nanoparticles	Crystal violet	633 nm	10^7	41
Dimer of Ag nanoparticles	Rhodamine 6G	500 nm	1.4×10^9	42
Ag nanoparticles	4-aminobenzenethiol	632 nm	5.0×10^5	43
Ag shell – Au satellite (Ag – Au SS)	4-fluorobenzenethiol	785 nm	1.4×10^6	44
bimetallic (Au/Ag) hierarchical structure	2-naphthalenthio	633 nm	2×10^7	45
Au@Ag core-shell nanocubes	4-Mercaptobenzoic acid	785 nm	2.2×10^6	46
Nanoporous gold nanoframes	4-methylbenzenethiol	785 nm	$\sim 10^4$	23
silver core-gold shell nanostructures	alkanethiol, 11- mercaptoundecanoic acid	633 nm	6.51×10^5	47
Ag@Au nanowire	4-nitrothiophenol	532 nm	1.3×10^9	48
N-th nanoframes	2-Naphthalenthio	633 nm	1.3×10^9	This work

3) Another concern is about the composition of the nanoframes. The composition of the nanoframes play an important role in SERS. Here, EDS was introduced to determine the atomic ratio of Au, Ag, and Pt in nanoframes. As we know, EDS is useful in analyzing the composition of nanostructured materials, however, do not have sufficient accuracy. Other characterizations, such as ICP-MS, are required to reveal the atomic ratios of different elements in "N-th nanoframes".

Answer: As following the reviewer's comment, we conducted ICP-MS analysis and calculated the atomic ratio of different elements in "N-th nanoframes". We added the plotted data to Supplementary Fig. S14 and changed the sentence from "Typically, Ag atomic fractions of 1st, 2nd, 3rd, and 4th of Ag nanoframes were 35%, 30%, 42%, and 30%, respectively." to "Typically, through the ICP-MS analysis, Ag atomic fractions of 1st, 2nd, 3rd, and 4th of Ag nanoframes were found to be 22%, 20%, 19%, and 20%, respectively (Supplementary Fig. 14)." in manuscript (page 7)

	1 st Ag nanoframes	2 nd Ag nanoframes	3 rd Ag nanoframes	4 th Ag nanoframes
Pt content (%)	51	48	44	51
Au content (%)	27	32	37	29
Ag content (%)	22	20	19	20

Collectively, this work provides a potential route for the construction of complex nanostructures to obtain highly-sensitive SERS materials for chemical detection. However, the weakness in the SERS part strongly limited the importance of the work, and the matters mentioned above concern the key idea and conclusions of the paper; hence it is recommended for publication after a major revision.

Reviewer #4 (Remarks to the Author):

In the manuscript "N-th nanoframe", Sungjae Yoo, Jaewon Lee, Hajir Hilal, Insub Jung, Woongkyu Park, Joong Wook Lee, Soobong Choi, and Sungho Park describe the synthesis of multi-walled nanocages from polyhedra. Despite a small lack of details, the synthesis is elaborately performed and matches state of the art colloidal synthesis and finds itself in the field of physical chemistry and plasmonics well. However, as already obvious by the too minimalistic title, novelty, uniqueness, and a groundbreaking application are missing and I don't see much advantages as compared to other nanorattles, nanocages, and especially nanomatryoshkas, which are well known in literature. Exploiting Pt frames as bars and the selective deposition of silver seems like a novel and smart way of fabrication, but I think it is too specialized for the broad readership of Nature Communications, but rather would fit in a journal focusing on physical chemistry.

Independently, the following points should be addressed, clarified, and discussed:

1. In what physical sense are these nanocages 4 dimensional?

Answer: We think our complex nanoframes have a high degree of structural controllability in a single entity analogous to 4-dimensional nanoframes that can modulate their structural features as function of time. But, actually, it is difficult to mention that our complex nanoframes are 4 dimensional structure and thus, we removed the word "4-dimensional nanoframes" in our manuscript.

2. The thin Ag layer enabling faceted and epitaxial growth needs to be proven by HR-TEM directly showing this layer and alternative mechanisms, like UPD, need to be ruled out.

For the presented EDX mappings (e.g. Fig S3 and S10) the signal-to-noise ratio is too low in order to proof a homogeneous and epitaxial growth. Further, the EDX spectra should be shown in the supporting information, especially since the signals of Pt(M) and Au(M) are close, as well as those of Ag(L) and Cl(K), and thus, do not allow clear separation in all cases.

Answer: As following the reviewer's comment, we added EDX spectra of Au@Pt NPs and typically, 4th Ag nanoframes in supplementary Fig. 3 and 13. As shown in below figures, signals of Pt, Au and Ag could be clearly observed.

For Au@Pt

For 4th Ag nanoframes

It is difficult to obtain the HR TEM image showing the thin Ag layer on the Au@Pt nanoparticles. Additionally, to prove the formation of the thin Ag layer indirectly, we conducted experiment that in the well-faceted overgrowth step, the amount of Ag⁺ ions was controlled while retaining other parameters constant. As shown in below figures, the surface of the resulting Au nanostructures is gradually faceted as the amount of Ag⁺ ions increases in a reaction solution. Taken together, it can be expected that with relatively small amount of Ag⁺ ions (Ag⁺/Au³⁺ = 0.05, 0.1, and 0.25), thin Ag layer could not be formed uniformly on Pt sites of Au@Pt nanoparticles resulting in the island growth pattern of Au, but, with relatively large amount of Ag⁺ ions (Ag⁺/Au³⁺ = 0.5, 1, and 2), thin Ag layer could be fully covered on Pt regions generating the well-faceted growth of Au.

3. Is the inner frame fixed in position or is it freely movable in the outer frame? (e.g. Fig S6 looks collapsed)

If it is fixed, how is it fixed and how do you guarantee stable fixation? How does this fixation influence the plasmonics, as most probably it will happen on the smallest distance between the rims?

If it is free, how is a collapse of the well-defined structure and plasmonic hotspots avoided and ensured that the rims are separated during the final silver overgrowth?

Answer: Inner frame is freely movable in the outer frame because there is no physical connector after inner Au etching step which results in inner frame attached on side of outer frame. We think that Ag atoms were uniformly deposited on whole surface area, demonstrating that the rims were separated during Ag growth step as shown in Supplementary Fig. 13B.

4. How thick and homogeneous is the silver shell for the SERS measurements? HR-TEM (and EDX) are inevitable, also to exclude roughness as source of field enhancement.

Answer: To investigate thickness and morphology of Ag shell on the Pt nanoframes, we conducted TEM and EDX analysis for 1st Pt nanoframes and Ag nanoframes as shown in below figures. In comparison with figures A to E which show the morphology and element distribution of Pt nanoframes, Figures F to J show that Ag atoms were smoothly coated on the whole surface of Pt nanoframes after depositing Ag step. Thickness of Ag shell was found to be ~ 5 nm based on zoomed-in TEM image (Figure H) and EDX image mapping data (Figure I). From these results, it can be proved that effect of roughness in a field enhancement the is almost negligible.

5. The optical properties of all particle stages should be discussed and shown, especially in regard to the highly damping Pt. How thick, does the silver shell around those bars need to be, in order to dominate the plasmonics?

Answer: As following the reviewer' comment, we added UV-vis spectrum showing optical properties of all particle stages to supplementary Fig. 5, 6, and 9, and added the sentences saying "Shape transformation from 1st-Au-O-NPs to 2nd -Au@Pt-O-NPs could be monitored through UV-vis spectrum (Supplementary Fig. 5). After rim-selective growth of Pt, LSPR peaks red-shifted accompanying with peak damping due to enlarging the size of nanoparticle and poor plasmonic activity of Pt in visible regions. In contrast, LSPR peaks blue-shifted accompanying with increased extinction after well-faceted overgrowth of Au. This is because the resulting nanostructures act as pure Au nanoparticles." to manuscript (Page 5), "Shape transformation from 1st-Au-TO-NPs to 3rd-Au@Pt-O-NPs could be also monitored through UV-vis spectrum (Supplementary Fig. 6)." to manuscript (Page 5), and "Shape transformation from 1st-Au-C-NPs to 4th-Au@Pt-O-NPs could be also monitored through UV-vis spectrum (Supplementary Fig. 9)". to manuscript (Page 6)

In order to investigate the plasmonic properties of Ag nanoframes as function of the amount of Ag coating on the core Pt nanoframes, we controlled Ag shell thickness through controlling the amount of Ag ions in a reaction solution. As shown in below SEM images and corresponding UV-vis-NIR spectrum, Pt nanoframes with thickness of 9 nm (Figure A) showed no plasmonic band. Upon depositing Ag on the Pt nanoframes, plasmonic band started to appear at 1111 nm. As the thickness of Ag shell increases further, the plasmonic band blue-shifted from 1100 nm (for thickness of 3 nm) to 1073 nm (for thickness of 5 nm), and 869 nm (for thickness of 7 nm). Taken together, it could be demonstrated that Ag thickness of 5 nm is required to appear the dominant plasmonic peak.

6. As plasmonics are the main application of these particles, an in-depth discussion of the resulting plasmonic modes should be included and a (hybridization) model explaining their nature should be proposed.

Answer: We obtained the theoretical calculated UV-vis-NIR spectrum for Ag nanoframes with different number of nanoframes through FEM method as shown in below Figure. We discussed and added the sentence saying "Dipole mode of 1st Ag nanoframes blue-shifted from 936 nm to 534 nm resulting from surface plasmon coupling between inner and outer nanoframe. Then, as the number of nanoframes increased from double to quadruple, dipole mode of nanoframes red-shifted from 534 nm to 564 nm and 620 nm accompanying with peak broadening. This trend well agreed with theoretical calculated UV-vis-NIR spectrum obtained by theoretical simulations based on a finite-element method (FEM) as shown in Supplementary Fig. 15. Charge distribution shown in Figure 5 (red: electron rich, blue: electron deficient) clearly indicates that the surface charge density significantly increased in a single entity as the number of nanoframes increased from single to quadruple, which is a distinctive characteristic of N-th nanoframes. We expect that surface plasmon coupling would occur in a complex manner, representing a broad LSPR feature (Fig. 5A)." to manuscript (page 7)

7. As the laser for the SERS measurements matches best to higher order frames, how is excluded, that this is a main source of the increased intensity?

Answer: We added single particle SERS signals of Ag nanoframes with different number of frames using 532 and 785 nm laser. With 532 laser, Raman signals increases as the number of frames increases, as shown in below Figures. However, with 785 nm laser, Raman signal could not be appeared. Taken together, among three different excitation wavelengths, the well resolved Raman signals could be observed at 633 nm excitation wavelength as shown Supplementary Fig. 18.

We added data showing reproducibility of SERS signals at 532 and 785 nm excitation wavelength to Supplementary Fig. 17 and 19, respectively and changed the sentence from "In addition, SERS signals are obtained by measuring 40 single particles, indicating highly uniform and reproducible SERS signals (Supplementary Fig. S16)." to "In addition, SERS signals are obtained at different excitation wavelength, indicating that the highest intensity of Raman signals with relatively high reproducibility SERS signals was observed at the 633 nm excitation wavelength (Supplementary Fig. S17, 18, and 19).

At 532 nm laser excitation,

At 785 nm laser excitation

REVIEWER COMMENTS

Reviewer #1 (Remarks to the Author):

The revised version of the manuscript represents a major improvement. I would like to thank the authors for considering the comments and suggestions, and especially for their effort in revising the experimental section.

I am satisfied with their answers, and I recommend the manuscript for publication in Nature Communications.

Reviewer #2 (Remarks to the Author):

The authors have satisfactorily addressed all comments. I recommend publication as is.

Reviewer #3 (Remarks to the Author):

The authors have fully addressed my concerns.

The enhancement factors of N-th nanoframes have been calculated, which is $\sim 1.3 \times 10^9$ for 4th Ag nanoframes.

ICP-MS analyses reveal that 1st, 2nd, 3rd, and 4th of Ag nanoframes possess similar Ag atomic fraction ($\sim 20\%$).

A table has been added to compare the SERS enhancement factor of N-th nanoframe with other reported SERS substrates.

From my point of view, the paper can be published in its revised form.

Reviewer #4 (Remarks to the Author):

The authors significantly improved the overall quality of the manuscript and were able to close several gaps in the line of argumentation.

However, in my opinion, I still don't see much practical advantages as compared to other nanorattles, nanocages, and especially nanomatryoshkas - the authors did not comment on this either directly to the referees or in the manuscript.

Consequently, the authors could not convince me, that it has the impact for the broad readership of Nature Communications, but would fit much better in a specialized physical chemistry journal.

Reviewer #4 (Remarks to the Author):

The authors significantly improved the overall quality of the manuscript and were able to close several gaps in the line of argumentation.

However, in my opinion, I still don't see much practical advantages as compared to other nanorattles, nanocages, and especially nanomatryoshkas - the authors did not comment on this either directly to the referees or in the manuscript.

Consequently, the authors could not convince me, that it has the impact for the broad readership of nature communications, but would fit much better in a specialized physical chemistry journal.

Answer: We appreciate the reviewer's critical comments on the benefits of N-th nanoframes as compared to nanomatryoshkas. We believe that the current synthetic strategy for N-th nanoframes is unique in terms of their controllability within one entity. The resulting morphology is in between the closed (i.e. shells) and open (i.e. single frames) systems. It is feasible to utilize the inner nanostructures for further enhancing their corresponding optical and physical properties, which is in a sharp contrast to other similar systems (i.e. nanorattles, nanocages, and nanomatryoshkas). For example, although nanorattles and nanomatryoshkas can generate the structural synergistic effects among nanostructures within one entity, outer shell of those structures inhibits penetration of chemical molecules into interior part. Therefore, there is limitation in direct detection of chemical species. Alternatively, if N-th nanoframes are employed as a building block and form three-dimensional superstructures, electromagnetic near fields will be a lot enhanced through an additional inter-particles near field coupling (undergoing research). In addition, the nanoporous morphology will allow the efficient trapping of chemical or biological molecules inside of the superstructure, where the electromagnetic near fields are highly amplified, leading to highly sensitive SERS detection. This effect might be huge for detecting the extremely low concentration of unknown chemical or biological molecules. Although we show their near-field focusing as a function of the number of wrapping frames, we believe there are many other related applications by using N-th nanoframes.

We added a sentence in manuscript (page 8) to explain the high potentials of N-th nanoframes in various applications. It is saying, "The synthetic strategy for N-th nanoframes is unique in terms of their controllability within one entity. The resulting morphology consists of overlapped nanoframes with each different shape, which is in between the open frame structure and closed shell structure systems. It is feasible to utilize the inner nanostructures for further enhancing their corresponding optical and physical properties, which differs from other similar systems (e.g. nanorattles, nanocages, and nanomatryoshkas). Although we show one example at the current stage, their near-field focusing as a function of the number of wrapping nanoframes, we believe that there are many other related applications by using N-th nanoframes, especially wherein the inner morphology control, penetration depth of light, and mass-transportation of analytes are important."